# Deep Compression Autoencoder for Efficient High-Resolution Diffusion Models

**Junyu Chen**[1,2*], **Han Cai**[3*†], **Junsong Chen**[3], **Enze Xie**[3],
**Shang Yang**[1], **Haotian Tang**[1], **Muyang Li**[1], **Song Han**[1,3]
[1]MIT   [2]Tsinghua University   [3]NVIDIA
https://github.com/mit-han-lab/efficientvit

## Abstract

We present Deep Compression Autoencoder (DC-AE), a new family of autoencoders for accelerating high-resolution diffusion models. Existing autoencoders have demonstrated impressive results at a moderate spatial compression ratio (e.g., $8\times$), but fail to maintain satisfactory reconstruction accuracy for high spatial compression ratios (e.g., $64\times$). We address this challenge by introducing two key techniques: (1) **Residual Autoencoding**, where we design our models to learn residuals based on the space-to-channel transformed features to alleviate the optimization difficulty of high spatial-compression autoencoders; (2) **Decoupled High-Resolution Adaptation**, an efficient decoupled three-phase training strategy for mitigating the generalization penalty of high spatial-compression autoencoders. With these designs, we improve the autoencoder's spatial compression ratio up to 128 while maintaining the reconstruction quality. Applying our DC-AE to latent diffusion models, we achieve significant speedup without accuracy drop. For example, on ImageNet $512 \times 512$, our DC-AE provides **19.1×** inference speedup and **17.9×** training speedup on H100 GPU for UViT-H while achieving a better FID, compared with the widely used SD-VAE-f8 autoencoder.

## 1 Introduction

Latent diffusion models (Rombach et al., 2022) have emerged as a leading framework and demonstrated great success in image synthesis (Labs, 2024; Esser et al., 2024). They employ an autoencoder to project the images to the latent space to reduce the cost of diffusion models. For example, the predominantly adopted solution in current latent diffusion models (Rombach et al., 2022; Labs, 2024; Esser et al., 2024; Chen et al., 2024b;a) is to use an autoencoder with a spatial compression ratio of 8 (denoted as f8), which converts images of spatial size $H \times W$ to latent features of spatial size $\frac{H}{8} \times \frac{W}{8}$. This spatial compression ratio is satisfactory for low-resolution image synthesis (e.g., $256 \times 256$). However, for high-resolution image synthesis (e.g., $1024 \times 1024$), further increasing the spatial compression ratio is critical, especially for diffusion transformer models (Peebles & Xie, 2023; Bao et al., 2023) that have quadratic computational complexity to the number of tokens.

The current common practice for further reducing the spatial size is downsampling on the diffusion model side. For example, in diffusion transformer models (Peebles & Xie, 2023; Bao et al., 2023), this is achieved by using a patch embedding layer with patch size $p$ that compresses the latent features to $\frac{H}{8p} \times \frac{W}{8p}$ tokens. In contrast, little effort has been made on the autoencoder side. The main bottleneck hindering the employment of high spatial-compression autoencoders is the reconstruction accuracy drop. For example, Figure 2 (a) shows the reconstruction results of SD-VAE (Rombach et al., 2022) on ImageNet $256 \times 256$ with different spatial compression ratios. We can see that the rFID (reconstruction FID) degrades from 0.90 to 28.3 if switching from f8 to f64.

This work presents **Deep Compression Autoencoder (DC-AE)**, a new family of high spatial-compression autoencoders for efficient high-resolution image synthesis. By analyzing the underlying source of the accuracy degradation between high spatial-compression and low spatial-

---

[*]Equal contribution. Junyu Chen is an intern at MIT during this work.
[†]Project lead. Correspondence to: hcai@nvidia.com, songhan@mit.edu.

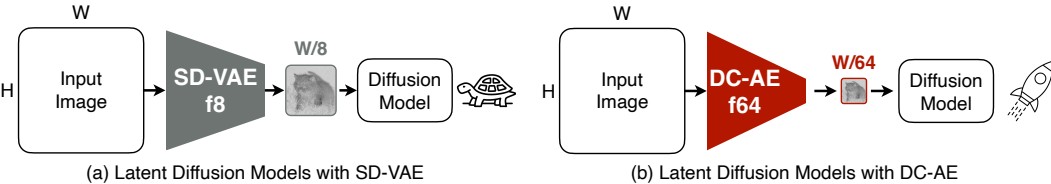

Figure 1: DC-AE accelerates diffusion models by increasing autoencoder's spatial compression ratio.

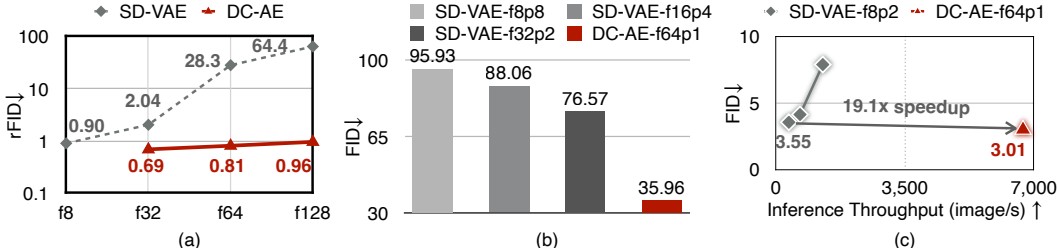

Figure 2: **(a) Image Reconstruction Results on ImageNet 256×256.** f denotes the spatial compression ratio. When the spatial compression ratio increases, SD-VAE has a significant reconstruction accuracy drop (higher rFID) while DC-AE does not have this issue. **(b) ImageNet 512×512 Image Generation Results on UViT-S with Various Autoencoders.** p denotes the patch size. Shifting the token compression task to the autoencoder enables the diffusion model to focus more on the denoising task, leading to better FID. **(c) Comparison to SD-VAE-f8 on ImageNet 512×512 with UViT Variants.** DC-AE-f64p1 provides 19.1× higher inference throughput and 0.54 better ImageNet FID than SD-VAE-f8p2 on UViT-H.

compression autoencoders, we find high spatial-compression autoencoders are more difficult to optimize (Section 3.1) and suffer from the generalization penalty across resolutions (Figure 3 b). To this end, we introduce two key techniques to address these two challenges. First, we propose **Residual Autoencoding** (Figure 4) to alleviate the optimization difficulty of high spatial-compression autoencoders. It introduces extra non-parametric shortcuts to the autoencoder to let the neural network modules learn residuals based on the space-to-channel operation. Second, we propose **Decoupled High-Resolution Adaptation** (Figure 6) to tackle the other challenge. It introduces a high-resolution latent adaptation phase and a low-resolution local refinement phase to avoid the generalization penalty while maintaining a low training cost.

With these techniques, we increase the spatial compression ratio of autoencoders to 32, 64, and 128 while maintaining good reconstruction accuracy (Table 2). The diffusion models can fully focus on the denoising task with our DC-AE taking over the whole token compression task, which delivers better image generation results than prior approaches (Table 3). For example, replacing SD-VAE-f8 with our DC-AE-f64, we achieve **17.9×** higher H100 training throughput and **19.1×** higher H100 inference throughput on UViT-H (Bao et al., 2023) while improving the ImageNet $512 \times 512$ FID from 3.55 to 3.01. We summarize our contributions as follows:

- We analyze the challenges of increasing the spatial compression ratio of autoencoders and provide insights into how to address these challenges.

- We propose Residual Autoencoding and Decoupled High-Resolution Adaptation that effectively improve the reconstruction accuracy of high spatial-compression autoencoders, making their reconstruction accuracy feasible for use in latent diffusion models.

- We build DC-AE, a new family of autoencoders based on our techniques. It delivers significant training and inference speedup for diffusion models compared with prior autoencoders.

## 2 RELATED WORK

**Autoencoder for Diffusion Models.** Training and evaluating diffusion models directly in high-resolution pixel space results in prohibitive computational costs. To address this issue, Rombach et al. (2022) proposes latent diffusion models that operate in a compressed latent space produced

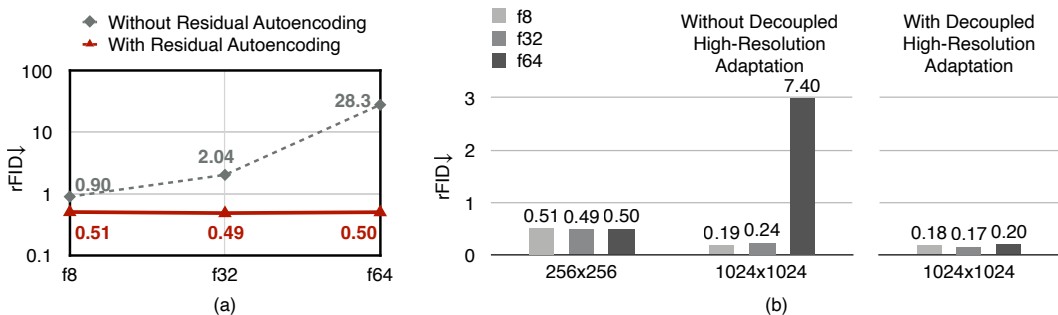

Figure 3: (a) High spatial-compression autoencoders are more difficult to optimize. Even with the same latent shape and stronger learning capacity, it still cannot match the f8 autoencoder's rFID. (b) High spatial-compression autoencoders suffer from significant reconstruction accuracy drops when generalizing from low-resolution to high-resolution.

by pretrained autoencoders. The proposed autoencoder with $8\times$ spatial compression ratio and $4$ latent channels has been widely adopted in subsequent works (Peebles & Xie, 2023; Bao et al., 2023). Since then, follow-up works mainly focus on enhancing the reconstruction accuracy of the f8 autoencoder by increasing the number of latent channels (Esser et al., 2024; Dai et al., 2023; Labs, 2024). Additionally, to improve the reconstruction quality, Zhu et al. (2023) leverages a heavier decoder and incorporates task-specific priors. In contrast to prior works, our work focuses on an orthogonal direction, increasing the spatial compression ratio of the autoencoders (e.g., f64). To the best of our knowledge, our work is the first study in this critical but underexplored direction.

**Diffusion Model Acceleration.** Diffusion models have been widely used for image generation and showed impressive results (Labs, 2024; Esser et al., 2024). However, diffusion models are computationally intensive, motivating many works to accelerate diffusion models. One representative strategy is reducing the number of inference sampling steps by training-free few-step samplers (Song et al., 2021; Lu et al., 2022a;b; Zheng et al., 2023; Zhang & Chen, 2023; Zhang et al., 2023; Zhao et al., 2024b; Shih et al., 2024; Tang et al., 2024) or distilling-based methods (Meng et al., 2023; Salimans & Ho, 2022; Yin et al., 2024b;a; Song et al., 2023; Luo et al., 2023; Liu et al., 2023). Another representative strategy is model compression by leveraging sparsity (Li et al., 2022; Ma et al., 2024b) or quantization (He et al., 2024; Fang et al., 2024; Li et al., 2023; Zhao et al., 2024a). Designing efficient diffusion model architectures (Li et al., 2024d; Liu et al., 2024; Cai et al., 2024) or inference systems (Li et al., 2024b; Wang et al., 2024) is also an effective approach for boosting efficiency. In addition, improving the data quality (Chen et al., 2024b;a) can boost the training efficiency of diffusion models.

All these works focus on diffusion models while the autoencoder remains the same. Our work opens up a new direction for accelerating diffusion models, which can benefit both training and inference.

## 3 METHOD

In this section, we first analyze why existing high spatial-compression autoencoders (e.g., SD-VAE-f64) fail to match the accuracy of low spatial-compression autoencoders (e.g., SD-VAE-f8). Then we introduce our Deep Compression Autoencoder (DC-AE) with *Residual Autoencoding* and *Decoupled High-Resolution Adaptation* to close the accuracy gap. Finally, we discuss the applications of our DC-AE to latent diffusion models.

### 3.1 MOTIVATION

We conduct ablation study experiments to get insights into the underlying source of the accuracy gap between high spatial-compression and low spatial-compression autoencoders. Specifically, we consider three settings with gradually increased spatial compression ratio, from f8 to f64.

Each time the spatial compression ratio increases, we stack additional encoder and decoder stages upon the current autoencoder. In this way, high spatial-compression autoencoders contain low spatial-compression autoencoders as sub-networks and thus have higher learning capacity.

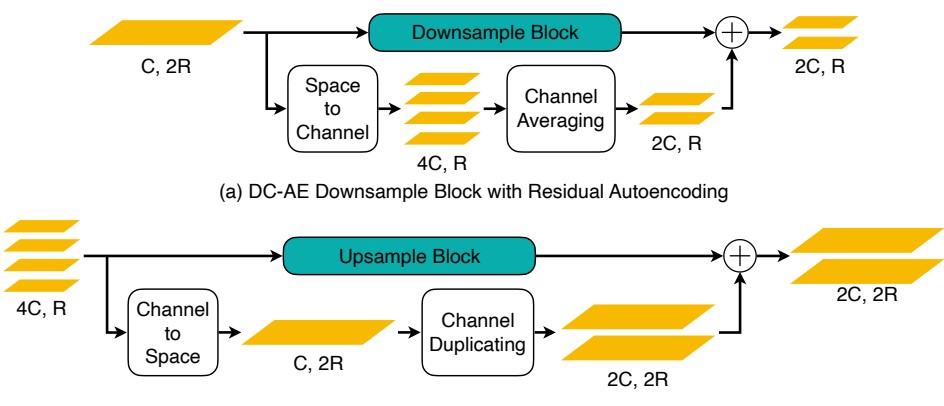

Figure 4: **Illustration of Residual Autoencoding.** It adds non-parametric shortcuts to let the neural network modules learn residuals based on the space-to-channel operation. 'C' denotes the number of channels. 'R' denotes the image size.

Additionally, we increase the latent channel number to maintain the same total latent size across different settings. We can then convert the latent to a higher spatial compression ratio one by applying a space-to-channel operation (Shi et al., 2016): $H \times W \times C \to \frac{H}{p} \times \frac{W}{p} \times p^2 C$.

We summarize the results in Figure 3 (a, gray dash line). Even with the same total latent size and stronger learning capacity, we still observe degraded reconstruction accuracy when the spatial compression ratio increases. It demonstrates that *the added encoder and decoder stages (consisting of multiple SD-VAE building blocks) work worse than a simple space-to-channel operation.*

Based on this finding, we conjecture *the accuracy gap comes from the model learning process: while we have good local optimums in the parameter space, the optimization difficulty hinders high spatial-compression autoencoders from reaching such local optimums.*

### 3.2 DEEP COMPRESSION AUTOENCODER

**Residual Autoencoding.** Motivated by the analysis, we introduce Residual Autoencoding to address the accuracy gap. The general idea is depicted in Figure 4. The core difference from the conventional design is that we explicitly let neural network modules learn the downsample residuals based on the space-to-channel operation to alleviate the optimization difficulty. Different from ResNet (He et al., 2016), the residual here is not identity mapping, but space-to-channel mapping.

In practice, this is implemented by adding extra non-parametric shortcuts on the encoder's downsample blocks and decoder's upsample blocks. Specifically, for the downsample block, the non-parametric shortcut is a space-to-channel operation followed by a non-parametric channel averaging operation to match the channel number. For example, assuming the downsample block's input feature map shape is $H \times W \times C$ and its output feature map shape is $\frac{H}{2} \times \frac{W}{2} \times 2C$, then the added shortcut is:

$$H \times W \times C \xrightarrow{\text{space-to-channel}} \frac{H}{2} \times \frac{W}{2} \times 4C$$
$$\xrightarrow{\text{split into two groups}} \underbrace{[\frac{H}{2} \times \frac{W}{2} \times 2C, \frac{H}{2} \times \frac{W}{2} \times 2C] \xrightarrow{\text{average}} \frac{H}{2} \times \frac{W}{2} \times 2C.}_{\text{channel averaging}}$$

Accordingly, for the upsample block, the non-parametric shortcut is a channel-to-space operation followed by a non-parametric channel duplicating operation:

$$\frac{H}{2} \times \frac{W}{2} \times 2C \xrightarrow{\text{channel-to-space}} H \times W \times \frac{C}{2}$$
$$\xrightarrow{\text{duplicate}} \underbrace{[H \times W \times \frac{C}{2}, H \times W \times \frac{C}{2}] \xrightarrow{\text{concat}} H \times W \times C.}_{\text{channel duplicating}}$$

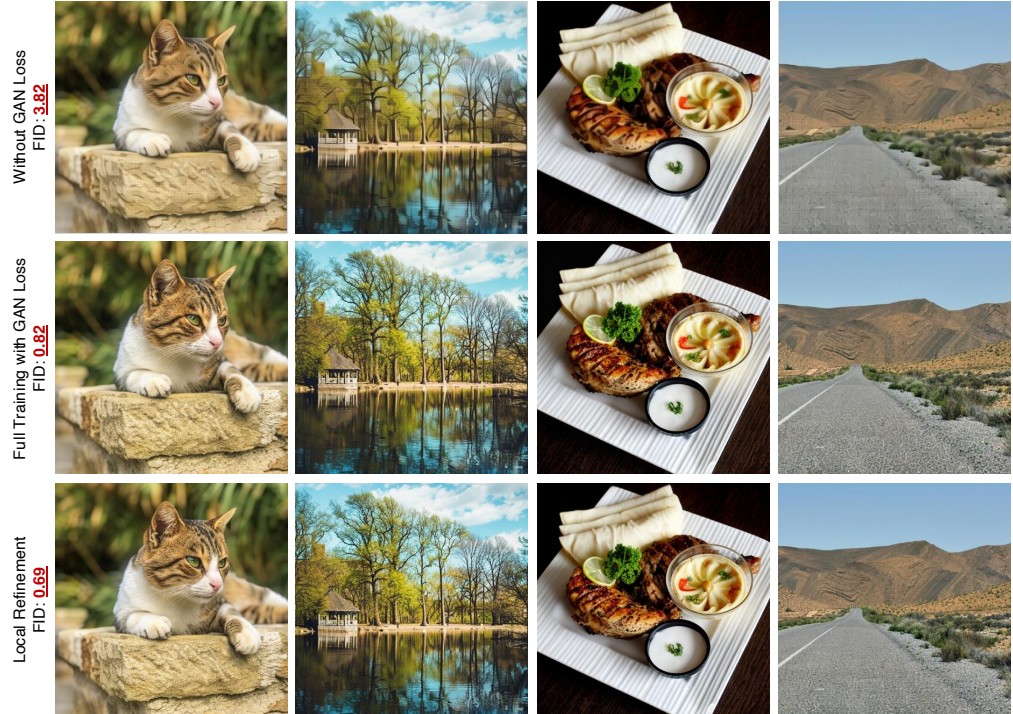

Figure 5: Autoencoder already learns to reconstruct content and semantics without GAN loss, while GAN loss improves local details and removes local artifacts. We replace the GAN loss full training with lightweight local refinement training which achieves the same goal and has lower training cost.

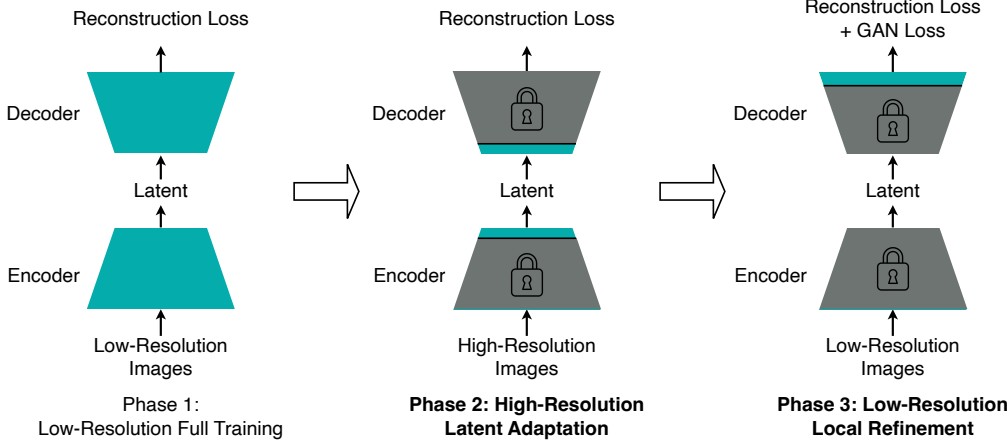

Figure 6: **Illustration of Decoupled High-Resolution Adaptation.**

In addition to the downsample and upsample blocks, we also change the middle stage design following the same principle (Figure 10 b, right).

Figure 3 (a) shows the comparison with and without our Residual Autoencoding on ImageNet $256 \times 256$. We can see that Residual Autoencoding effectively improves the reconstruction accuracy of high spatial-compression autoencoders.

**Decoupled High-Resolution Adaptation.**    Residual Autoencoding alone can address the accuracy gap when handling low-resolution images. However, when extending it to high-resolution images, we find it not sufficient. Due to the large cost of high-resolution training, the common practice for high-resolution diffusion models is directly using autoencoders trained on low-resolution images (e.g., $256 \times 256$) (Chen et al., 2024b;a). This strategy works well for low spatial-compression

**ImageNet 512×512 (Class-Conditional)**

| Diffusion Model | Autoencoder | Patch Size | #Tokens | FID (w/o CFG) ↓ | FID (w/ CFG) ↓ |
|---|---|---|---|---|---|
| UViT-S [1] | SD-VAE-f8 | 8 | 64 | 125.08 | 95.93 |
| | SD-VAE-f16 | 4 | 64 | 115.32 | 88.06 |
| | SD-VAE-f32 | 2 | 64 | 107.33 | 76.57 |
| | DC-AE-f64 | 1 | 64 | **67.30** | **35.96** |

Table 1: **Ablation Study on Patch Size and Autoencoder's Spatial Compression Ratio.**

autoencoders. However, high spatial-compression autoencoders suffer from a significant accuracy drop. For example, in Figure 3 (b), we can see that f64 autoencoder's rFID degrades from 0.50 to 7.40 when generalizing from $256 \times 256$ to $1024 \times 1024$. In contrast, the f8 autoencoder's rFID improves from 0.51 to 0.19 under the same setting. Additionally, we also find this issue more severe when using a higher spatial compression ratio. In this work, we refer to this phenomenon as the *generalization penalty of high spatial-compression autoencoders*. A straightforward solution to address this issue is conducting training on high-resolution images. However, it suffers from a large training cost and unstable high-resolution GAN loss training.

We introduce Decoupled High-Resolution Adaptation to tackle this challenge. Figure 6 demonstrates the detailed training pipeline. Compared with the conventional single-phase training strategy (Rombach et al., 2022), our Decoupled High-Resolution Adaptation has two key differences.

First, we decouple the GAN loss training from the full model training and introduce a dedicated local refinement phase for the GAN loss training. In the local refinement phase (Figure 6, phase 3), we only tune the head layers of the decoder while freezing all the other layers. The intuition of this design is based on the finding that the reconstruction loss alone is sufficient for learning to reconstruct the content and semantics. Meanwhile, the GAN loss mainly improves local details and removes local artifacts (Figure 5). Achieving the same goal of local refinement, only tuning the decoder's head layers has a lower training cost and delivers better accuracy than the full training.

Moreover, the decoupling prevents the GAN loss training from changing the latent space. This approach enables us to conduct the local refinement phase on low-resolution images without worrying about the generalization penalty. This further reduces the training cost of phase 3 and avoids the highly unstable high-resolution GAN loss training.

Second, we introduce an additional high-resolution latent adaptation phase (Figure 6, phase 2) that tunes the middle layers (i.e., encoder's head layers and decoder's input layers) to adapt the latent space for alleviating the generalization penalty. In our experiments, we find only tuning middle layers is sufficient for addressing this issue (Figure 3 b) while having a lower training cost than high-resolution full training (memory cost: 153.98 GB → 67.81 GB)[1] (Cai et al., 2020).

### 3.3 APPLICATION TO LATENT DIFFUSION MODELS

Applying our DC-AE to latent diffusion models is straightforward. The only hyperparameter to change is the patch size (Peebles & Xie, 2023). For diffusion transformer models (Peebles & Xie, 2023; Bao et al., 2023), increasing the patch size $p$ is the common approach for reducing the number of tokens. It is equivalent to first applying the space-to-channel operation to reduce the spatial size of the given latent by $p \times$ and then using the transformer model with a patch size of 1.

Since combining a low spatial-compression autoencoder (e.g., f8) with the space-to-channel operation can also achieve a high spatial compression ratio, a natural question is how it compares with directly reaching the target spatial compression ratio with DC-AE.

We conduct ablation study experiments and summarize the results in Table 1. We can see that directly reaching the target spatial compression ratio with the autoencoder gives the best results among all settings. In addition, we also find that shifting the spatial compression ratio from the diffusion model to the autoencoder consistently leads to better FID.

---

[1]Assuming the input resolution is $1024 \times 1024$ and the batch size is 12.

| ImageNet 256×256 | Latent Shape | Autoencoder | rFID ↓ | PSNR ↑ | SSIM ↑ | LPIPS ↓ |
|---|---|---|---|---|---|---|
| f32c32 | 8×8×32 | SD-VAE [40] | 2.64 | 22.13 | 0.59 | 0.117 |
| | | DC-AE | **0.69** | **23.85** | **0.66** | **0.082** |
| f64c128 | 4×4×128 | SD-VAE [40] | 26.65 | 18.07 | 0.41 | 0.283 |
| | | DC-AE | **0.81** | **23.60** | **0.65** | **0.087** |
| ImageNet 512×512 | Latent Shape | Autoencoder | rFID ↓ | PSNR ↑ | SSIM ↑ | LPIPS ↓ |
| f64c128 | 8×8×128 | SD-VAE [40] | 16.84 | 19.49 | 0.48 | 0.282 |
| | | DC-AE | **0.22** | **26.15** | **0.71** | **0.080** |
| f128c512 | 4×4×512 | SD-VAE [40] | 100.74 | 15.90 | 0.40 | 0.531 |
| | | DC-AE | **0.23** | **25.73** | **0.70** | **0.084** |
| FFHQ 1024×1024 | Latent Shape | Autoencoder | rFID ↓ | PSNR ↑ | SSIM ↑ | LPIPS ↓ |
| f64c128 | 16×16×128 | SD-VAE [40] | 6.62 | 24.55 | 0.68 | 0.237 |
| | | DC-AE | **0.23** | **31.04** | **0.83** | **0.061** |
| f128c512 | 8×8×512 | SD-VAE [40] | 179.71 | 18.11 | 0.63 | 0.585 |
| | | DC-AE | **0.41** | **31.18** | **0.83** | **0.062** |
| MapillaryVistas 2048×2048 | Latent Shape | Autoencoder | rFID ↓ | PSNR ↑ | SSIM ↑ | LPIPS ↓ |
| f64c128 | 32×32×128 | SD-VAE [40] | 7.55 | 22.37 | 0.68 | 0.262 |
| | | DC-AE | **0.36** | **29.57** | **0.84** | **0.075** |
| f128c512 | 16×16×512 | SD-VAE [40] | 152.09 | 17.82 | 0.67 | 0.594 |
| | | DC-AE | **0.38** | **29.70** | **0.84** | **0.074** |

Table 2: **Image Reconstruction Results.**

## 4 EXPERIMENTS

### 4.1 SETUPS

**Implementation Details.** We use a mixture of datasets to train autoencoders (baselines and DC-AE), containing ImageNet (Deng et al., 2009), SAM (Kirillov et al., 2023), MapillaryVistas (Neuhold et al., 2017), and FFHQ (Karras et al., 2019). For ImageNet experiments, we exclusively use the ImageNet training split to train autoencoders and diffusion models. The model architecture is similar to SD-VAE (Rombach et al., 2022) except for our new designs discussed in Section 3.2. In addition, we use the original autoencoders instead of the variational autoencoders for our models, as they perform the same in our experiments and the original autoencoders are simpler. We also replace transformer blocks with EfficientViT blocks (Cai et al., 2023) to make autoencoders more friendly for handling high-resolution images while maintaining similar accuracy.

For image generation experiments, we apply autoencoders to diffusion transformer models including DiT (Peebles & Xie, 2023) and UViT (Bao et al., 2023). We follow the same training settings as the original papers. Additionally, we build USiT by combining UViT (Bao et al., 2023) with the SiT sampler (Ma et al., 2024a). The SiT and USiT models are trained for 500k iterations with batch size 1024. We consider three settings with different resolutions, including ImageNet (Deng et al., 2009) for $512 \times 512$ generation, FFHQ (Karras et al., 2019) and MJHQ (Li et al., 2024a) for $1024 \times 1024$ generation, and MapillaryVistas (Neuhold et al., 2017) for $2048 \times 2048$ generation.

**Efficiency Profiling.** We profile the training and inference throughput on the H100 GPU with PyTorch and TensorRT respectively. The latency is measured on the 3090 GPU with batch size 2. The training memory is profiled using PyTorch, assuming a batch size of 256. We use fp16 for all cases.

### 4.2 IMAGE COMPRESSION AND RECONSTRUCTION

Table 2 summarizes the results of DC-AE and SD-VAE (Rombach et al., 2022) under various settings (f represents the spatial compression ratio and c denotes the number of latent channels). DC-AE provides significant reconstruction accuracy improvements than SD-VAE for all cases. For example,

| Diffusion Model | Autoencoder | Patch Size | NFE | Throughput (image/s) ↑ Training | Inference | Latency (ms) ↓ | Memory (GB) ↓ | FID ↓ w/o CFG | w/ CFG |
|---|---|---|---|---|---|---|---|---|---|
| DiT-XL [38] | Flux-VAE-f8 [20] | 2 | 250 | 54 | 0.83 | 7915 | 56.3 | 27.35 | 8.72 |
| | Asym-VAE-f8 [58] | 2 | 250 | 54 | 0.85 | 7686 | 56.2 | 11.39 | 2.97 |
| | SD-VAE-f8 [40] | 2 | 250 | 54 | 0.85 | 7686 | 56.2 | 12.03 | 3.04 |
| | DC-AE-f32 | 1 | 250 | **241** | **4.03** | **1958** | **20.9** | 9.56 | 2.84 |
| | DC-AE-f32‡ | 1 | 250 | **241** | **4.03** | **1958** | **20.9** | **6.88** | **2.41** |
| UViT-H [1] | Flux-VAE-f8 [20] | 2 | 30 | 55 | 5.82 | 913 | 54.2 | 30.91 | 12.63 |
| | Asym-VAE-f8 [58] | 2 | 30 | 55 | 5.85 | 914 | 54.1 | 11.36 | 3.51 |
| | SD-VAE-f8 [40] | 2 | 30 | 55 | 5.85 | 914 | 54.1 | 11.04 | 3.55 |
| | DC-AE-f32 | 1 | 30 | 247 | 27.03 | 246 | 18.6 | **9.83** | **2.53** |
| | DC-AE-f64 | 1 | 30 | **984** | **111.77** | **104** | **10.6** | 13.96 | 3.01 |
| | DC-AE-f64† | 1 | 30 | **984** | **111.77** | 105 | **10.6** | 12.26 | 2.66 |
| UViT-2B [1] | Asym-VAE-f8 [58] | 2 | 30 | 27 | 2.62 | 2243 | OOM | 9.87 | 3.62 |
| | SD-VAE-f8 [40] | 2 | 30 | 27 | 2.62 | 2243 | OOM | 9.73 | 3.57 |
| | DC-AE-f32 | 1 | 30 | 112 | 11.08 | 590 | 42.0 | 8.13 | 2.30 |
| | DC-AE-f64 | 1 | 30 | **450** | **45.55** | **258** | **30.2** | 7.78 | 2.47 |
| | DC-AE-f64† | 1 | 30 | **450** | **45.55** | **258** | **30.2** | **6.50** | **2.25** |
| MAGVIT-v2 [51] | - | - | - | - | - | - | - | 3.07 | 1.91 |
| EDM2-XXL [17] | - | - | - | - | - | - | - | **1.91** | 1.81 |
| MAR-L [24] | - | - | - | - | - | - | - | 2.74 | 1.73 |
| SiT-XL [33] | DC-AE-f32 | 1 | - | 241 | - | - | 20.9 | 7.47 | 2.41 |
| USiT-H | DC-AE-f32 | 1 | - | 247 | - | - | 18.6 | 3.80 | 1.89 |
| USiT-2B | DC-AE-f32 | 1 | - | 112 | - | - | 42.0 | 2.90 | **1.72** |

Table 3: **Class-Conditional Image Generation Results on ImageNet 512×512.** † represents the model is trained for 4× training iterations (i.e., 500K → 2,000K iterations). ‡ represents the model is trained with 4× batch size (i.e., 256 → 1024). 'NFE' denotes the number of functional evaluations. The NFEs for SiT (Ma et al., 2024a) and USiT models are left blank as they use an adaptive-step evaluation scheduler.

| Diffusion Model | Autoencoder | Patch Size | NFE | Throughput (image/s) ↑ Training | Inference | Latency (ms) ↓ | Memory (GB) ↓ | MJHQ 512×512 FID ↓ | CLIP Score ↑ |
|---|---|---|---|---|---|---|---|---|---|
| PIXART-α [6] | SD-VAE-f8 [40] | 2 | 20 | 43 | 7.81 | 742 | 60.45 | 6.3 | 26.36 |
| | DC-AE-f32 | 1 | 20 | **173** | **31.27** | **209** | **23.77** | **6.1** | **26.41** |

Table 4: **Text-to-Image Generation Results.**

on ImageNet $512 \times 512$, DC-AE improves the rFID from 16.84 to 0.22 for the f64c128 autoencoder and 100.74 to 0.23 for the f128c512 autoencoder.

In addition to the quantitative results, Figure 7 shows image reconstruction samples produced by SD-VAE and DC-AE. Reconstructed images by DC-AE demonstrate a better visual quality than SD-VAE's reconstructed images. In particular, for the f64 and f128 autoencoders, DC-AE still maintains a good visual quality for small text and the human face.

### 4.3 LATENT DIFFUSION MODELS

We compare DC-AE with the widely used SD-VAE-f8 autoencoder (Rombach et al., 2022) on various diffusion transformer models. For DC-AE, we always use a patch size of 1 (denoted as p1). For SD-VAE-f8, we follow the common setting and use a patch size of 2 or 4 (denoted as p2, p4). The results are summarized in Table 3, Table 4, and Figure 9.

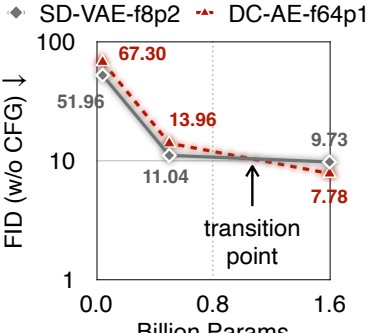

Figure 9: **Model Scaling Results on ImageNet 512×512 with UViT.** DC-AE-f64 benefits more from scaling up than SD-VAE-f8.

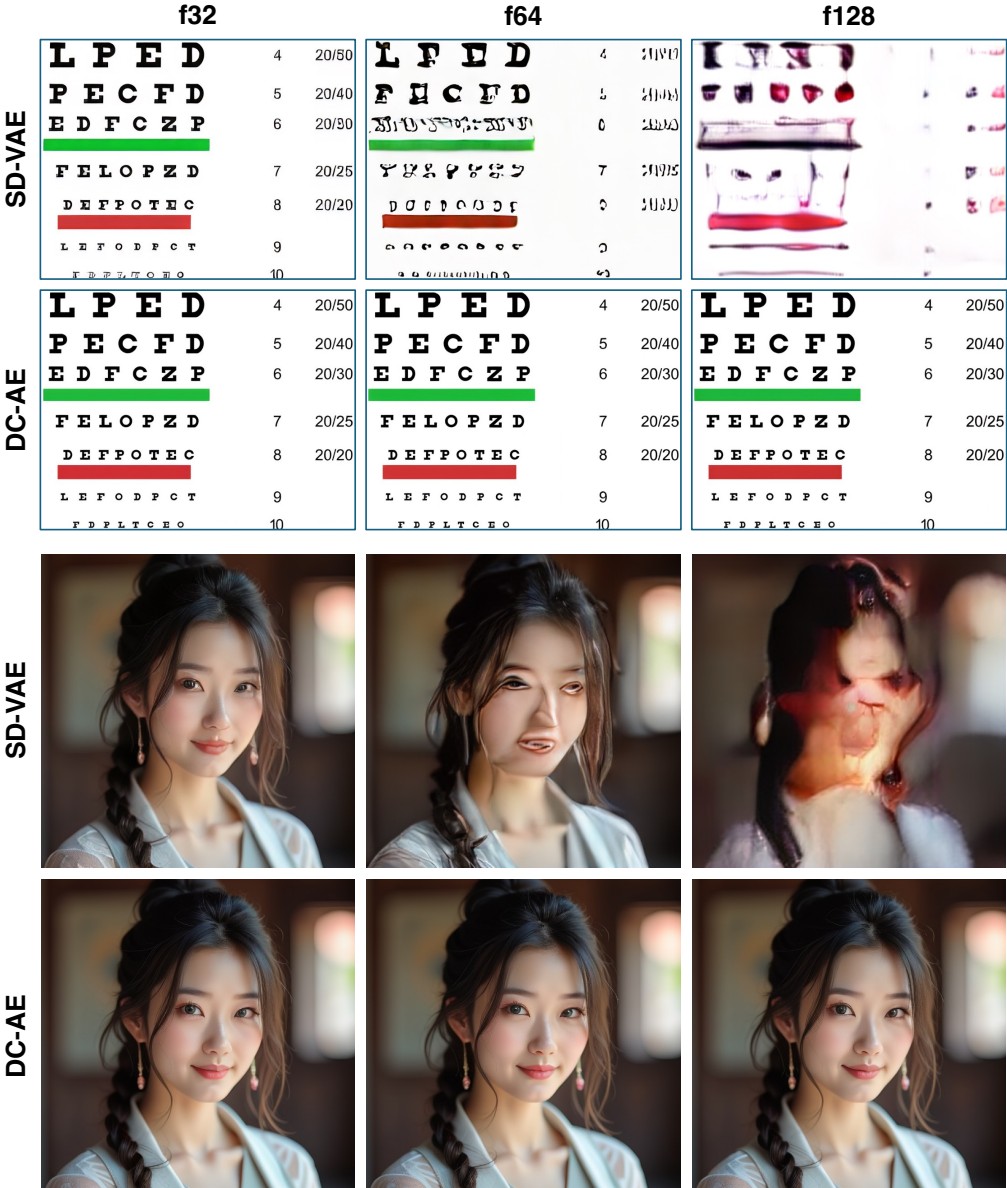

Figure 7: **Autoencoder Image Reconstruction Samples.**

**ImageNet 512×512.** As shown in Table 3, DC-AE-f32p1 consistently delivers better FID than SD-VAE-f8p2 on all diffusion transformer models. In addition, it has 4× fewer tokens than SD-VAE-f8p2, leading to 4.5× higher H100 training throughput and 4.8× higher H100 inference throughput for DiT-XL. We also observe that larger diffusion transformer models seem to benefit more from our DC-AE (Figure 9). For example, DC-AE-f64p1 has a worse FID than SD-VAE-f8p2 on UViT-S but a better FID on UViT-2B. We conjecture it is because DC-AE-f64 has a larger latent channel number than SD-VAE-f8, thus needing more model capacity (Esser et al., 2024).

Applying DC-AE to USiT models, we achieve highly competitive results compared with prior leading image generative models. For example, DC-AE-f32+USiT-2B achieves 1.72 FID on ImageNet 512×512, outperforming the SOTA diffusion model EDM2-XXL and SOTA auto-regressive image generative models (MAGVIT-v2 and MAR-L).

**Text-to-Image Generation.** Table 4 reports our text-to-image generation results. All models are trained for 100K iterations from scratch. Similar to prior cases, we observe DC-AE-f32p1 provides a better FID and a better CLIP Score than SD-VAE-f8p2. Figure 8 demonstrates samples generated

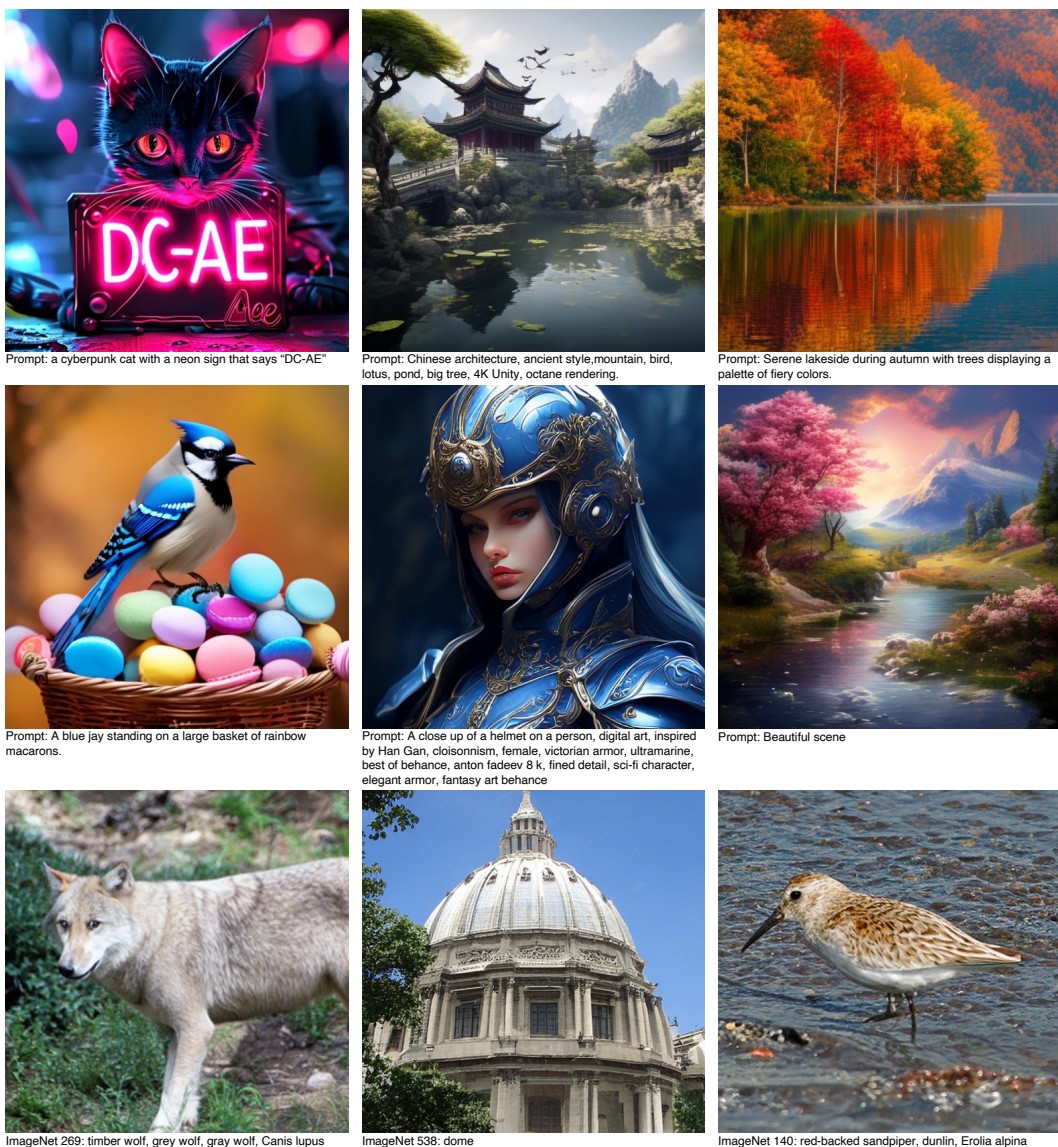

Prompt: a cyberpunk cat with a neon sign that says "DC-AE"

Prompt: Chinese architecture, ancient style,mountain, bird, lotus, pond, big tree, 4K Unity, octane rendering.

Prompt: Serene lakeside during autumn with trees displaying a palette of fiery colors.

Prompt: A blue jay standing on a large basket of rainbow macarons.

Prompt: A close up of a helmet on a person, digital art, inspired by Han Gan, cloisonnism, female, victorian armor, ultramarine, best of behance, anton fadeev 8 k, fined detail, sci-fi character, elegant armor, fantasy art behance

Prompt: Beautiful scene

ImageNet 269: timber wolf, grey wolf, gray wolf, Canis lupus

ImageNet 538: dome

ImageNet 140: red-backed sandpiper, dunlin, Erolia alpina

Figure 8: **Images Generated by Diffusion Model using Our DC-AE.**

by the diffusion models with our DC-AE, showing the capacity to synthesize high-quality images while being significantly more efficient than prior models.

## 5 CONCLUSION

We accelerate high-resolution diffusion models by designing deep compression autoencoders to reduce the number of tokens. We proposed two techniques: *residual autoencoding* and *decoupled high-resolution adaptation* to address the challenges brought by the high compression ratio. The resulting new autoencoder family DC-AE demonstrated satisfactory reconstruction accuracy with a spatial compression ratio of up to 128. DC-AE also demonstrated significant training and inference efficiency improvements when applied to latent diffusion models.

## ACKNOWLEDGEMENTS

We thank NVIDIA for donating the DGX machines. We thank MIT-IBM Watson AI Lab, MIT and Amazon Science Hub, MIT AI Hardware Program, and National Science Foundation for supporting this research.

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

# A    DC-AE Architecture and Training Details

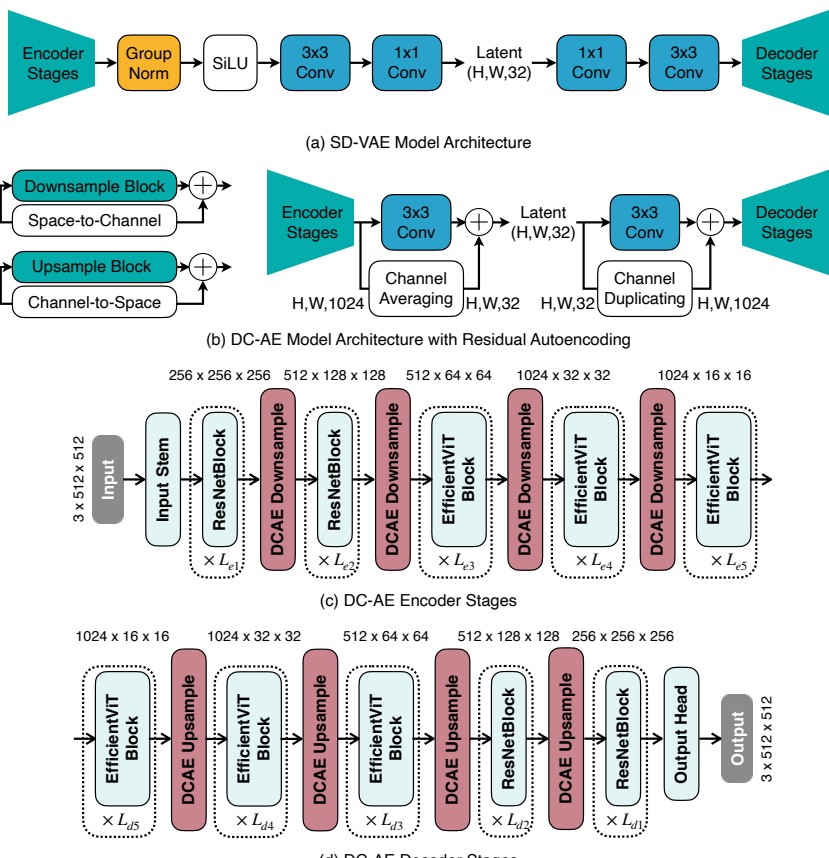

Figure 10: **Detailed Architecture of SD-VAE, DC-AE, DC-AE Encoder, and DC-AE Decoder Stages.**

We present the detailed architecture of SD-VAE, DC-AE, DC-AE encoder, and DC-AE decoder stages in Figure 10 to complement Figure 4.

We use the AdamW optimizer (Loshchilov, 2017) for all training phases.

In phase 1 (low-resolution full training), we use a constant learning rate of 6.4e-5 with a weight decay of 0.1, and AdamW betas of (0.9, 0.999). We use L1 loss and LPIPS loss (Zhang et al., 2018).

In phase 2 (high-resolution latent adaptation), we use a constant learning rate of 1.6e-5, a weight decay of 0.001, and AdamW betas of (0.9, 0.999). We use the same loss as phase 1.

In phase 3 (low-resolution local refinement), we use a constant learning rate of 5.4e-5, and AdamW betas of (0.5, 0.9). We use L1 loss, LPIPS loss (Zhang et al., 2018), and PatchGAN loss (Isola et al., 2017).

# B    Ablation Study on Training Different Numbers of Layers

Figure 11 presents the ablation study on training different numbers of layers in phase 2 (high-resolution latent adaptation) and phase 3 (low-resolution local refinement).

# C    Additional Image Reconstruction Results

Table 5 reports the reconstruction results under the low spatial-compression ratio setting. DC-AE delivers slightly better results than SD-VAE under this setting.

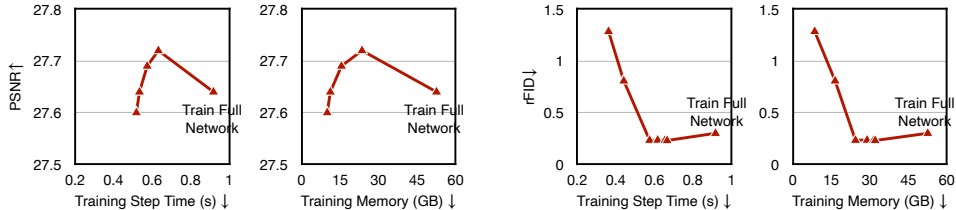

Figure 11: **Ablation Study on Training Different Numbers of Layers in Phase 2 (Left) and Phase 3 (Right).**

| **ImageNet 256×256** | Latent Shape | Autoencoder | rFID ↓ | PSNR ↑ | SSIM ↑ | LPIPS ↓ |
|---|---|---|---|---|---|---|
| f8c4 | 32×32×4 | SD-VAE [40] | 0.63 | 24.99 | 0.71 | 0.063 |
| | | DC-AE | **0.46** | **25.46** | **0.73** | **0.057** |

Table 5: **Image Reconstruction Results under the Low Spatial-Compression Ratio Setting.**

## D  LATENT SCALING AND SHIFTING FACTORS

Following the common practice (Rombach et al., 2022; Peebles & Xie, 2023; Bao et al., 2023; Esser et al., 2024; Labs, 2024; Chen et al., 2024b;a), we normalize the latent space of our autoencoders to apply to latent diffusion models. Given a dataset, we compute the root mean square of the latent features and use its multiplicative inverse as the scaling factor for our autoencoders. We do not use the shifting factor for our autoencoders.

## E  DIFFUSION MODEL ARCHITECTURE DETAILS

In addition to existing UViT models, we scaled the model up to 1.6B parameters, with a depth of 28, a hidden dimension of 2048, and 32 heads. We denote this model as UViT-2B.

## F  DIFFUSION SAMPLING HYPERPARAMETERS

For the DiT models, we use the DDPM (Ho et al., 2020) sampler from the DiT (Peebles & Xie, 2023) codebase with 250 sampling steps and a guidance scale of 1.3.

For the UViT models, we use the DPMSolver (Lu et al., 2022a) sampler with 30 sampling steps and a guidance scale of 1.5.

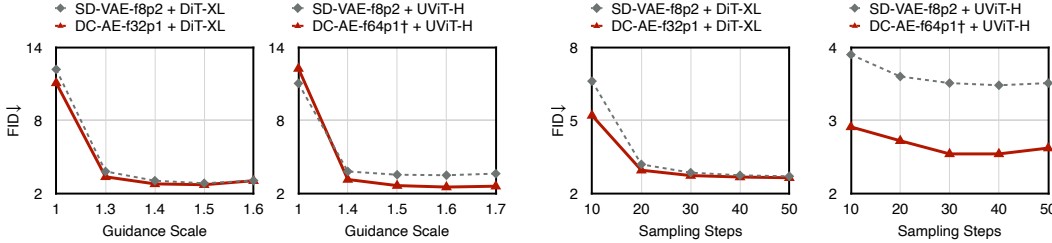

Figure 12: **Ablation Study on Diffusion Sampling Hyperparameters.** We use the DPMSolver sampler for both DiT-XL and UViT-H. DC-AE provides significant speedup over the baseline models while maintaining the generation performance under different diffusion sampling hyperparameters.

## G  HIGH-RESOLUTION IMAGE GENERATION RESULTS

Apart from ImageNet 512×512, we also test our models for higher-resolution image generation. As shown in Table 6, we have a similar finding where DC-AE-f32p1 achieves better FID than SD-VAE-f8p2 for all cases.

**FFHQ 1024×1024 (Unconditional) & MJHQ 1024×1024 (Class-Conditional)**

| Diffusion Model | Autoencoder | Patch Size | NFE | Throughput (image/s) ↑ Training | Inference | Latency (ms) ↓ | Memory (GB) ↓ | FFHQ FID ↓ w/o CFG | MJHQ FID ↓ w/o CFG | w/ CFG |
|---|---|---|---|---|---|---|---|---|---|---|
| DiT-S [38] | SD3-VAE-f8 [9] | 2 | 250 | 83 | 1.63 | 3554 | 41.4 | 46.28 | 109.43 | 103.02 |
| | Flux-VAE-f8 [20] | 2 | 250 | 83 | 1.63 | 3554 | 41.4 | 59.15 | 143.16 | 139.06 |
| | SDXL-VAE-f8 [39] | 2 | 250 | 84 | 1.67 | 3530 | 41.2 | 16.82 | 49.00 | 39.21 |
| | Asym-VAE-f8 [58] | 2 | 250 | 84 | 1.67 | 3530 | 41.2 | 17.10 | 48.30 | 38.35 |
| | SD-VAE-f8 [40] | 2 | 250 | 84 | 1.67 | 3530 | 41.2 | 16.98 | 48.05 | 38.19 |
| | | 4 | 250 | 470 | 11.13 | 632 | 10.7 | 23.81 | 60.94 | 51.29 |
| | DC-AE-f32 | 1 | 250 | 475 | 11.15 | 634 | 10.7 | 13.65 | 34.35 | 27.20 |
| | DC-AE-f32‡ | 1 | 250 | 475 | 11.15 | 634 | 10.7 | **11.39** | **28.36** | **21.89** |
| | DC-AE-f64 | 1 | 250 | **2085** | **50.26** | **230** | **3.1** | 26.88 | 61.30 | 53.38 |

**MapillaryVistas 2048×2048 (Unconditional)**

| Diffusion Model | Autoencoder | Patch Size | NFE | Throughput (image/s) ↑ Training | Inference | Latency (ms) ↓ | Memory (GB) ↓ | MapillaryVistas FID ↓ w/o CFG |
|---|---|---|---|---|---|---|---|---|
| DiT-S [38] | SD-VAE-f8 [40] | 4 | 250 | 84 | 1.64 | 3561 | 41.4 | 69.50 |
| | DC-AE-f64 | 1 | 250 | **459** | **10.91** | **639** | **11.0** | **59.55** |

Table 6: **1024×1024 and 2048×2048 Image Generation Results.** ‡ represents the model is trained with 4× batch size (i.e., 256 → 1024).

# H  IMAGE GENERATION RESULTS WITH OTHER EVALUATION METRICS

Table 7 presents a comprehensive evaluation of different diffusion models and autoencoders on ImageNet 512×512. The evaluation metrics include FID (Martin et al., 2017), inception score (IS) (Salimans et al., 2016), precision, recall (Kynkäänniemi et al., 2019), and CMMD (Jayasumana et al., 2024). Our DC-AE consistently delivers significant efficiency improvements while maintaining the generation performance under different evaluation metrics.

# I  ADDITIONAL SAMPLES

In Figure 13 and 14, we provide additional image reconstruction samples produced by SD-VAE and DC-AE. Reconstructed images by DC-AE demonstrate better visual qualities than SD-VAE's reconstructed images, especially for the f64 and f128 autoencoders. Some samples are cropped for better visualization of details like human faces and small texts.

In Figure 15 and Figure 16, we show randomly generated samples on ImageNet 512×512 and MJHQ-30K 512×512 by the diffusion models using our DC-AE.

| Diffusion Model | Autoencoder | Patch Size | NFE | Inference Throughput | FID↓ | | Inception Score↑ | | Precision↑ | | Recall↑ | | CMMD↓ | |
|---|---|---|---|---|---|---|---|---|---|---|---|---|---|---|
| | | | | | w/o CFG | w/ CFG | w/o CFG | w/ CFG | w/o CFG | w/ CFG | w/o CFG | w/ CFG | w/o CFG | w/ CFG |
| UViT-S [1] | SD3-VAE-f8 [9] | 2 | 30 | 49.73 | 164.34 | 143.82 | 6.07 | 7.53 | 0.06 | 0.09 | 0.31 | 0.39 | 3.13 | 2.94 |
| | Flux-VAE-f8 [20] | 2 | 30 | 49.73 | 106.07 | 84.73 | 13.39 | 17.71 | 0.28 | 0.37 | 0.39 | 0.42 | 1.90 | 1.67 |
| | SDXL-VAE-f8 [39] | 2 | 30 | 49.85 | 51.03 | 26.38 | 27.58 | 56.72 | 0.57 | 0.74 | 0.58 | 0.50 | 1.35 | 1.05 |
| | Asym-VAE-f8 [58] | 2 | 30 | 49.85 | 52.68 | 25.14 | 30.22 | 65.27 | 0.58 | 0.74 | 0.62 | 0.51 | 1.09 | 0.80 |
| | SD-VAE-f8 [40] | 2 | 30 | 49.85 | 51.96 | 24.57 | 30.37 | 65.73 | 0.57 | 0.74 | 0.64 | 0.52 | 1.23 | 0.91 |
| | SD-VAE-f16 [40] | 2 | 30 | 214.68 | 76.86 | 44.22 | 21.38 | 43.35 | 0.43 | 0.62 | 0.60 | 0.55 | 1.83 | 1.46 |
| | SD-VAE-f32 [40] | 1 | 30 | 214.72 | 70.23 | 38.63 | 23.07 | 47.72 | 0.46 | 0.64 | 0.58 | 0.56 | 1.71 | 1.36 |
| | DC-AE-f32 | 1 | 30 | 214.17 | 46.12 | 18.08 | 34.82 | 84.73 | 0.59 | 0.76 | 0.66 | 0.56 | 1.00 | 0.70 |
| | DC-AE-f64 | 1 | 30 | 896.23 | 67.30 | 35.96 | 24.55 | 52.86 | 0.44 | 0.64 | 0.60 | 0.56 | 1.44 | 1.14 |
| | DC-AE-f64† | 1 | 30 | 896.23 | 61.84 | 30.63 | 27.28 | 61.76 | 0.47 | 0.67 | 0.63 | 0.56 | 1.35 | 1.04 |
| DiT-XL [38] | Flux-VAE-f8 [20] | 2 | 250 | 0.83 | 27.35 | 8.72 | 53.09 | 130.20 | 0.68 | 0.83 | 0.61 | 0.48 | 0.54 | 0.30 |
| | Asym-VAE-f8 [58] | 2 | 250 | 0.85 | 11.39 | 2.97 | 108.70 | 241.10 | 0.75 | 0.83 | 0.65 | 0.53 | 0.37 | 0.20 |
| | SD-VAE-f8 [40] | 2 | 250 | 0.85 | 12.03 | 3.04 | 105.25 | 240.82 | 0.75 | 0.84 | 0.64 | 0.54 | 0.43 | 0.25 |
| | DC-AE-f32 | 1 | 250 | 4.03 | 9.56 | 2.84 | 117.49 | 226.98 | 0.75 | 0.82 | 0.64 | 0.55 | 0.34 | 0.22 |
| | DC-AE-f32‡ | 1 | 250 | 4.03 | 6.88 | 2.41 | 141.07 | 263.56 | 0.76 | 0.82 | 0.63 | 0.56 | 0.29 | 0.18 |
| UViT-H [1] | Flux-VAE-f8 [20] | 2 | 30 | 5.82 | 30.91 | 12.63 | 56.72 | 127.93 | 0.64 | 0.76 | 0.59 | 0.49 | 0.50 | 0.31 |
| | Asym-VAE-f8 [58] | 2 | 30 | 5.85 | 11.36 | 3.51 | 124.24 | 249.21 | 0.75 | 0.82 | 0.61 | 0.53 | 0.32 | 0.20 |
| | SD-VAE-f8 [40] | 2 | 30 | 5.85 | 11.04 | 3.55 | 125.08 | 250.66 | 0.75 | 0.82 | 0.61 | 0.53 | 0.39 | 0.26 |
| | DC-AE-f32 | 1 | 30 | 27.03 | 9.83 | 2.53 | 121.91 | 255.07 | 0.76 | 0.83 | 0.65 | 0.54 | 0.34 | 0.20 |
| | DC-AE-f64 | 1 | 30 | 111.77 | 13.96 | 3.01 | 99.20 | 229.16 | 0.73 | 0.83 | 0.64 | 0.53 | 0.50 | 0.31 |
| | DC-AE-f64† | 1 | 30 | 111.77 | 12.26 | 2.66 | 109.20 | 239.82 | 0.73 | 0.82 | 0.67 | 0.57 | 0.43 | 0.27 |
| UViT-2B [1] | Flux-VAE-f8 [20] | 2 | 30 | 2.58 | 25.03 | 10.12 | 74.04 | 161.29 | 0.67 | 0.78 | 0.58 | 0.51 | 0.38 | 0.24 |
| | Asym-VAE-f8 [58] | 2 | 30 | 2.62 | 9.87 | 3.62 | 131.95 | 258.63 | 0.76 | 0.83 | 0.59 | 0.52 | 0.30 | 0.19 |
| | SD-VAE-f8 [40] | 2 | 30 | 2.62 | 9.73 | 3.57 | 132.86 | 260.50 | 0.76 | 0.83 | 0.59 | 0.52 | 0.37 | 0.24 |
| | DC-AE-f32 | 1 | 30 | 11.08 | 8.13 | 2.30 | 135.44 | 272.73 | 0.76 | 0.82 | 0.66 | 0.56 | 0.30 | 0.17 |
| | DC-AE-f64 | 1 | 30 | 45.55 | 7.78 | 2.47 | 138.11 | 280.49 | 0.77 | 0.84 | 0.63 | 0.54 | 0.35 | 0.22 |
| | DC-AE-f64† | 1 | 30 | 45.55 | 6.50 | 2.25 | 152.35 | 293.45 | 0.77 | 0.83 | 0.65 | 0.56 | 0.31 | 0.19 |
| MAGVIT-v2 [51] | - | - | - | - | 3.07 | 1.91 | 213.1 | 324.3 | - | - | - | - | - | - |
| EDM2-XXL [17] | - | - | - | - | 1.91 | 1.81 | - | - | - | - | - | - | - | - |
| MAR-L [24] | - | - | - | - | 2.74 | 1.73 | 205.2 | 279.9 | - | - | - | - | - | - |
| SiT-XL [33] | DC-AE-f32 | 1 | - | - | 7.47 | 2.41 | 131.37 | 237.71 | 0.77 | 0.82 | 0.65 | 0.58 | 0.36 | 0.23 |
| USiT-H | DC-AE-f32 | 1 | - | - | 3.80 | 1.89 | 174.58 | 252.35 | 0.78 | 0.82 | 0.64 | 0.60 | 0.24 | 0.18 |
| USiT-2B | DC-AE-f32 | 1 | - | - | 2.90 | 1.72 | 187.68 | 248.10 | 0.79 | 0.82 | 0.63 | 0.61 | 0.21 | 0.17 |

Table 7: **Class-Conditional Image Generation Results on ImageNet 512×512 with More Evaluation Metrics.** † represents the model is trained for 4× training iterations (i.e., 500K → 2,000K iterations). ‡ represents the model is trained with 4× batch size (i.e., 256 → 1024). 'NFE' denotes the number of functional evaluations. The NFEs for SiT (Ma et al., 2024a) and USiT models are left blank as they use an adaptive-step evaluation scheduler.

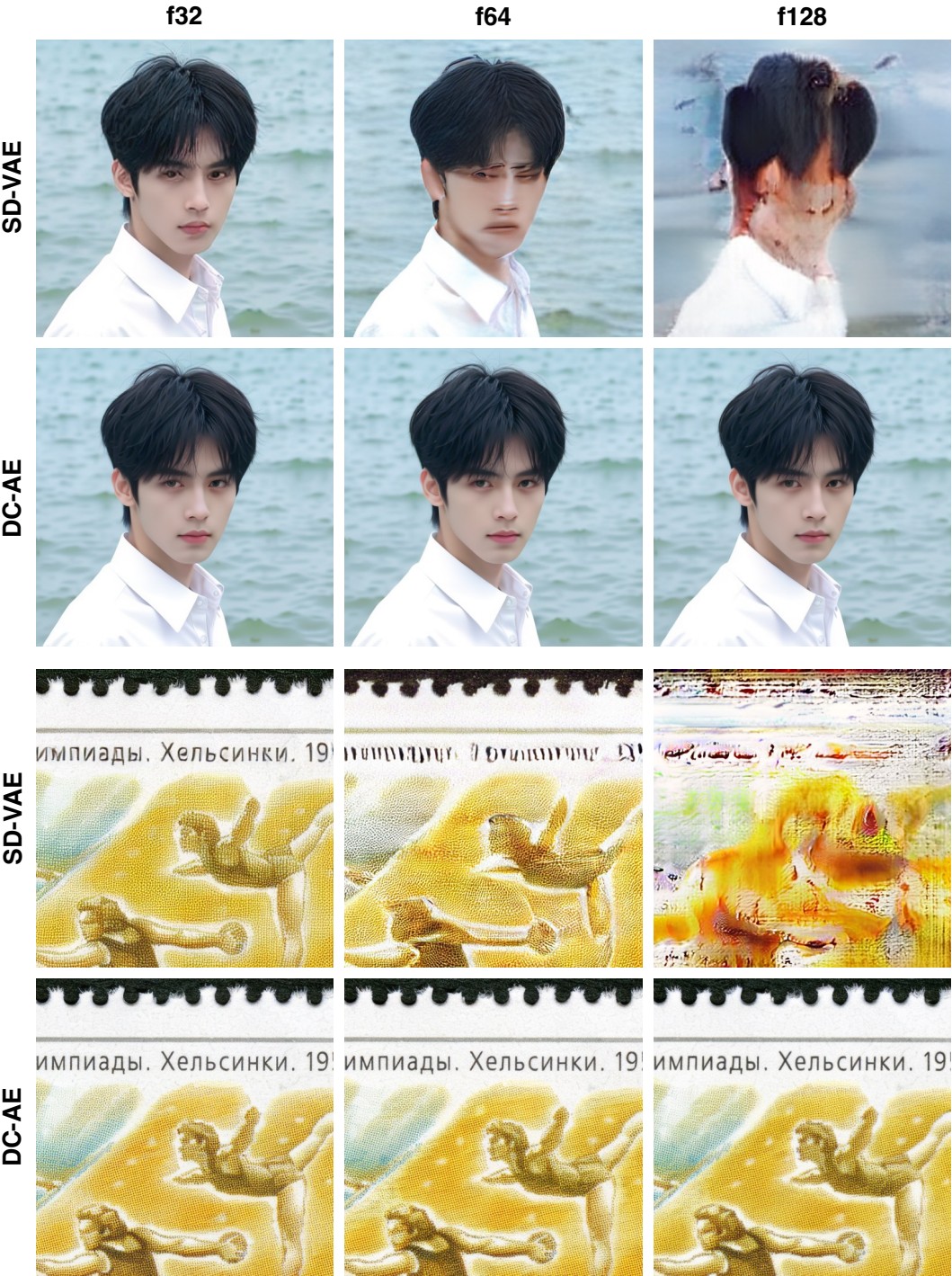

Figure 13: **Additional Autoencoder Image Reconstruction Samples.**

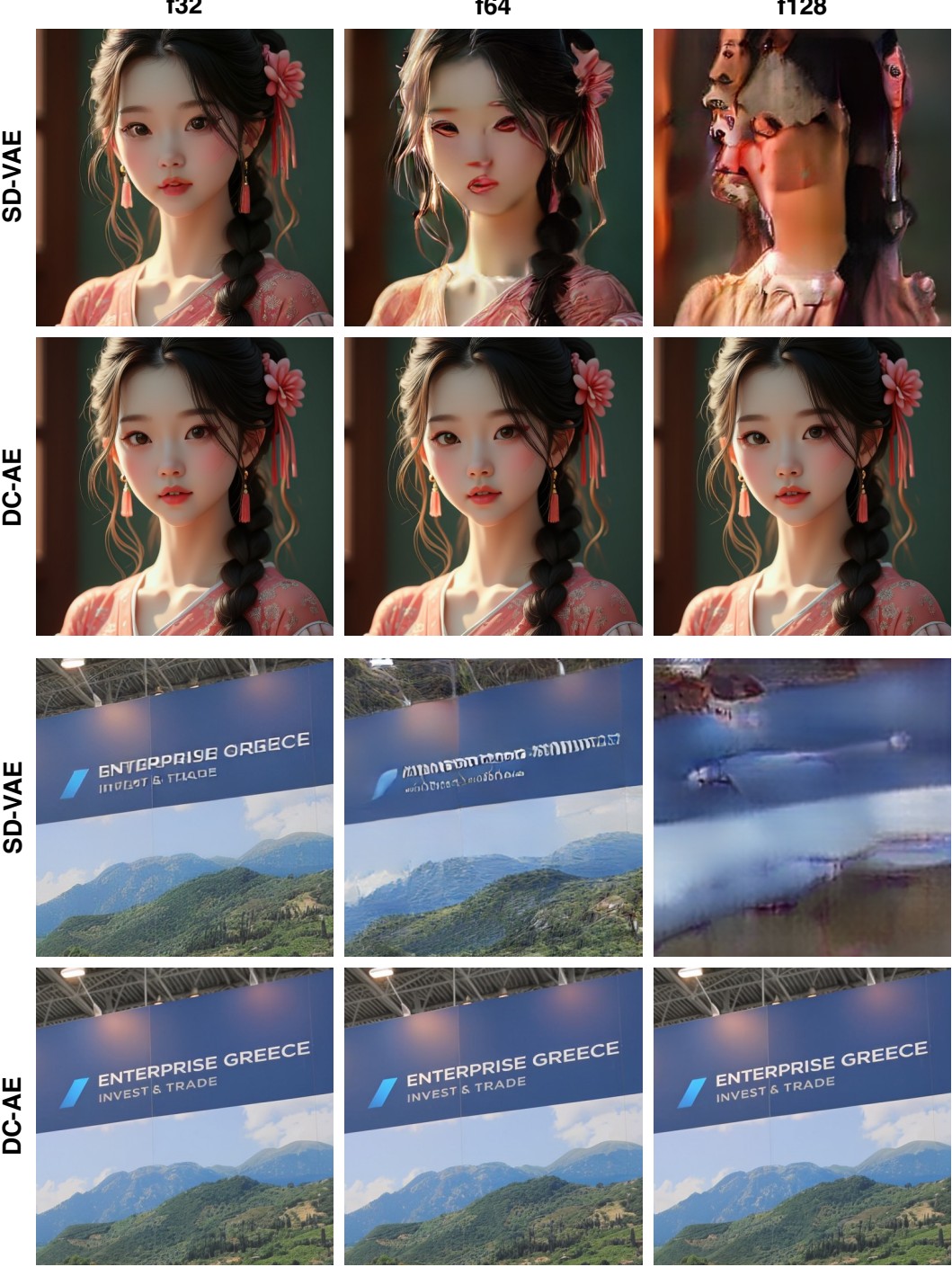

Figure 14: **Additional Autoencoder Image Reconstruction Samples.**

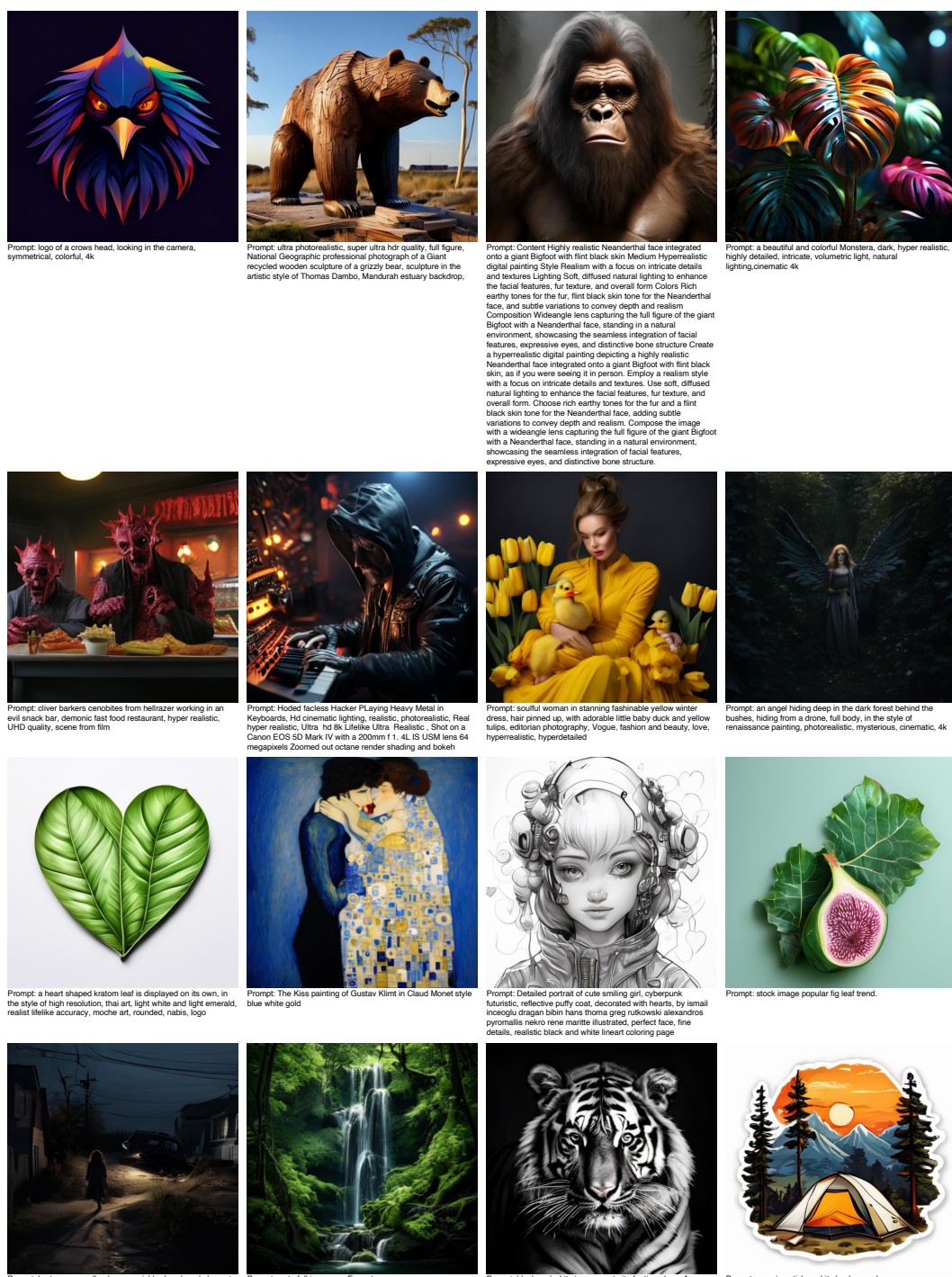

Figure 15: **Random 512×512 Text-to-Image Samples.** Prompts are randomly drawn from MJHQ-30K (Li et al., 2024a).

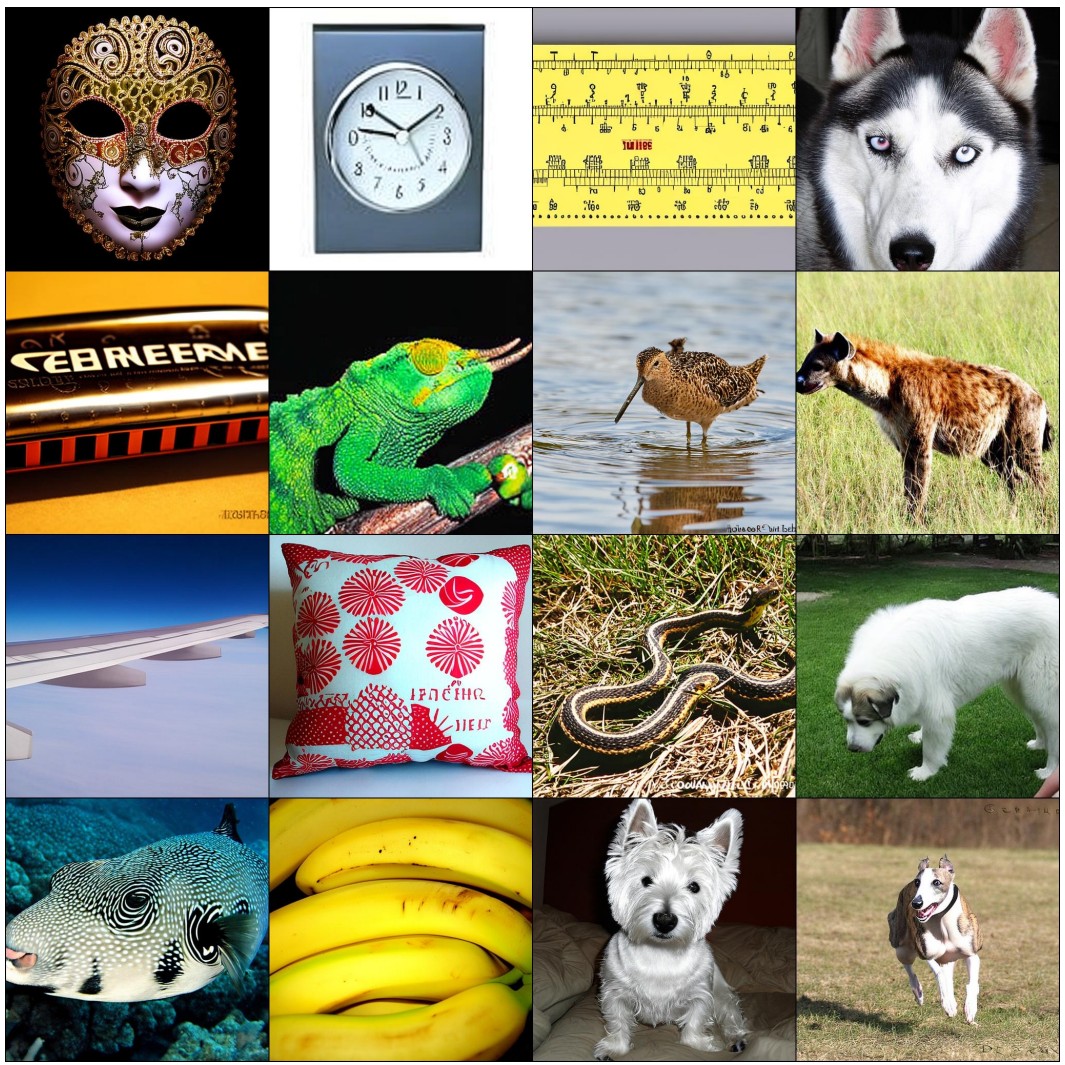

Figure 16: **Random Generated Samples on ImageNet 512×512.**

