# OpenReview forum: "Deep Compression Autoencoder for Efficient High-Resolution Diffusion Models"
_ICLR.cc/2025/Conference — ICLR 2025 Poster_

### Official Review · Reviewer_iQFp · 2024-11-02

**Soundness:** 3
**Presentation:** 2
**Contribution:** 3
**Rating:** 6
**Confidence:** 4

**Summary:**

The submission presents Deep Compression Autoencoder (DC-AE), a new series of autoencoders that enable highly efficient high-resolution image synthesis with latent diffusion models. The key architecture design is the adding residual connection with space-channel inter-play in both downsampling and upsampling layers, which alleviates the optimization difficulty significantly. They further propose decoupled high-resolution adaptation, a three-phase training strategy that mitigates the generalization penalty of high spatial-compression autoencoders. By improving the autoencoder's spatial compression ratio up to 128 while maintaining reconstruction quality, DC-AE provides significant speedups for both training and inference of latent diffusion models without sacrificing image generation accuracy. The authors demonstrate the effectiveness of DC-AE on various datasets and diffusion transformer architectures.

**Strengths:**

-	Novel autoencoder design for high compression ratios. The authors introduce additional non-parametric residual connections in the downsampling and upsampling layers, which facilitate the optimization of autoencoders in high spatial compression ratio scenarios.
-	The authors carefully design a three-stage training recipe to mitigate the high-resolution generalization gaps by fine-tuning the trained AE on high-resolution data with minimal additional costs.
-	The proposed DC-AEs show a substantial reduction in training and inference costs of latent diffusion models on various datasets.

**Weaknesses:**

-	Although the VAE shows little FID loss as it progresses to a higher spatial compression ratio (DC-AE-f32 to DC-AE-f64), it does not translate to similar results in the corresponding latent diffusion models as shown in Table 3. The underlying reason is probably that a diffusion model with higher capacity is required when dealing with high compression ratio latents.
-	The current submission lacks either theoretical analysis or illustrative toy examples on the proposed design choice of space-channel residual connection, e.g., comparisons on the loss landscapes.

**Questions:**

-	Could you please clarify the experimental settings and AE architectures in Section 3.1? By Line 161 ‘… convert the latent to higher spatial compression…’, do you mean that the f8 to f64 AEs have the same output latent size initially, yet f32 and f64 AEs further squeeze the space dimension into channel dimension?
-	In Table 5, the proposed method shows a lower CLIP score with the Pixart diffusion transformer, is it a typo?

---

> ### Author Response · Authors · 2024-11-25
> **Author Response**
>
> ### Q1: A diffusion model with higher capacity is required when dealing with the high compression ratio latent.
>
> Your hypothesis is correct. We further scaled the UViT model up to 1.6B parameters, with a depth of 28, a hidden dimension of 2048, and 32 heads. We denote this model as UViT-2B. We have added the results to **Appendix J (Table 9)** and included the model scaling results in **Appendix K (Figure 13)**. We can see that our DC-AE benefits more from scaling up the diffusion model than SD-VAE.
>
> ### Q2: Loss Landscape Comparison
> Thanks for the suggestion. We have added the loss landscape comparison between DC-AE and SD-VAE in **Appendix H (Figure 12)**. We find that DC-AE's loss landscape is flatter than SD-VAE's, indicating that it is easier for DC-AE to reach low-loss regions.
>
> ### Q3: Clarifications of Experimental Settings and AE Architectures in Section 3.1
> In Section 3.1, different autoencoders (f8, f32, f64) have the same total output latent size. For example, give a 1024x1024 image, f8's output latent size is 128x128x4 while f32's and f64's output latent sizes are 32x32x64 and 16x16x256 respectively. Therefore, f8's output latent can be converted to have the same shape as f32's or f64's output latent using a PixelUnshuffle (space-to-channel) operation.
>
> ### Q4: Typo in Table 5
> Thanks for pointing out the typo. We have fixed it in Table 5.

---

> > ### Comment · Reviewer_iQFp · 2024-12-02
> > **Thanks for the response**
> >
> > Thanks to the authors for the response. Most of my questions have been tackled, so I will retain my positive view of the submission. One remaining concern is the loss landscape. It seems from the figure that baseline SD-VAE has a better convergence rate. Maybe the authors can rescale the DC-VAE one for better visualization in the final revision.

---

### Official Review · Reviewer_pFgo · 2024-11-03

**Soundness:** 3
**Presentation:** 3
**Contribution:** 3
**Rating:** 6
**Confidence:** 4

**Summary:**

The paper proposes a new VAE architecture for latent diffusion generative modeling along with a multi-step training recipe. More specifically, the proposed design combines space-to-channels with averaging to build deep autoencoders with a high spatial subsampling factor (up to 64x). In order to train these models effectively at high resolution, they are first trained at low resolution without GAN loss, then partially tuned at higher resolution, and then refined with a GAN loss. The more aggressive subsampling leads to lower spatial resolution/sequence length in the latent space than VAEs from the literature and hence leads to substantial speed-ups of about 4x to 16x while maintaining generation quality, in combination with different latent diffusion models.

**Strengths:**

- The proposed architectural modifications are simple and sound, and the corresponding multi-step training recipe is relatively simple as well.
- The speed-ups are quite substantial.
- The method is tested on a broad range of data sets (including text-to-image modeling), and in combination with different latent diffusion models. So the results are quite comprehensive.

**Weaknesses:**

- The role of the space-to-channels-based autoencoder design and the multi-step training recipe are entangled. How does the baseline autoencoder (with more aggressive subsampling) perform, when it is trained with the multi-step recipe?
- The paper seems to lack important experimental details, including optimizer, learning rate, schedule etc. These are important in particular given the multi-stage training with different losses (including a GAN loss). Also, what losses does the method use exactly? It looks like the low resolution stage uses a perceptual loss (Figure 5).

\
Minor:
- In the side-by-side comparisons it would be useful to have the original image as additional reference.
- Some prior works discuss the relevance of space-to-channel in image enhancement/generation methods, probably worth mentioning (see e.g. Shi et al. "Real-time single image and video super-resolution using an efficient sub-pixel convolutional neural network." CVPR 2016)

**Questions:**

- What is the role of the averaging compared to simple space to channel? How would the model behave without the averaging? Would the channel dimension grow too fast?
- How does the use of EfficientViT blocks affect the quality (reconstruction/generation) of the method? Does it lead to better behavior when trained with large subsampling factors compared to the baseline?
- “..., we assume the number of sampling steps is 1.”: Does this also hold for inference throughput and latency? If so, I would suggest reporting the throughput/latency to sample a full image.
- What is the resolution of the images in Figure 7? It would generally be helpful to indicate the resolution for the visual examples in the caption.

---

> ### Author Response · Authors · 2024-11-25
> **Author Response**
>
> ### Q1: How does the baseline autoencoder (with more aggressive subsampling) perform, when it is trained with the multi-step recipe?
>
> We conduct ablation study experiments and summarize the results in **Appendix C (Table 6)**. Applying decoupled high-resolution adaptation to the baseline autoencoder improves the rFID from 16.84 to 5.54, still underperforming DC-AE.
>
> | Decoupled High-Resolution Adaptation | Residual Autoencoding | rFID  | PSNR  | SSIM | LPIPS |
> |:------------------------------------:|:---------------------:|:-----:|:-----:|:----:|:-----:|
> |                                      |                       | 16.84 | 19.49 | 0.48 | 0.282 |
> |               &check;                |                       | 5.54  | 21.13 | 0.54 | 0.228 |
> |               &check;                |         &check;       | **0.22**  | **26.15** | **0.71** | **0.080** |
>
> ### Q2: Experimental Details
>
> Thank you for the suggestion. We have added experimental details in **Appendix E**.
>
> ### Q3: Add the Original Image as an Additional Reference
>
> We have updated the figures, adding the original image as an additional reference, following your suggestion.
>
> ### Q4: Related Work
>
> Thank you for bringing the relevant works to our attention. We have revised our manuscript, adding citations to these works in Section 3.1.
>
> ### Q5: What is the role of the averaging compared to simple space to channel?
>
> We do not want the channel dimension to grow too fast. Therefore, we employ the averaging after the space-to-channel operation to match the target channel number.
>
> ### Q6: How does the use of EfficientViT blocks affect the quality of the method?
>
> The use of EfficientViT blocks is purely for making the autoencoder more friendly for tackling high-resolution images. The inference throughput comparison is summarized below. We can achieve a higher inference efficiency for the autoencoder on 1K and 4K images with the EfficientViT block.
>
> | Inference Throughput (samples/s) |     1024 |    4096  |
> | -------------------------------- |:--------:|:--------:|
> | EfficientViT                     | **4.42** | **0.26** |
> | Softmax Attention                |     2.22 |     0.02 |
>
> Regarding the reconstruction and generation quality, we do not observe a big difference between using the EfficientViT blocks and using the Softmax Attention blocks. In addition, it does not affect the inference efficiency of the latent diffusion models.
>
> | ImageNet 512x512 Reconstruction    | rFID | PSNR  | SSIM | LPIPS |
> | ---------------------------------- | :--: | :---: | :--: | :---: |
> | EfficientViT                       | 0.22 | 26.15 | 0.71 | 0.08  |
> | Softmax Attention                  | 0.22 | 26.14 | 0.71 | 0.08  |
>
> | ImageNet 512x512 Generation | Autoencoder       | FID (w/o CFG) | FID (w/ CFG) |
> | --------------------------- | ----------------- | :-----------: | :---------: |
> |                      UViT-S | EfficientViT      |         67.30 |         35.96 |
> |                      UViT-S | Softmax Attention |         67.79 |         36.49 |
> |                      UViT-H | EfficientViT      |         13.96 |          3.01 |
> |                      UViT-H | Softmax Attention |         13.83 |          2.96 |
>
> ### Q7: Report Throughput and Latency with Sampling Steps
>
> Thank you for your suggestion. We have updated the tables, adding an extra hyperparameter T to indicate the number of sampling steps.
>
> ### Q8: Resolution of the Images in Figure 7
>
> All the images in Figure 7 are reconstructed at resolution 1024x1024. The samples are cropped for better visualization of details like human faces and small texts. We have revised the caption of Figure 7 accordingly.

---

> > ### Comment · Reviewer_pFgo · 2024-12-01
> >
> > I thank the authors for their detailed rebuttal.
> >
> > I have one remaining question/suggestion (w.r.t. Q7). Do all the models use the same number of diffusion steps in Table 3, per section (or equivalently, per latent diffusion model architecture)? If not, I believe adding T would be somewhat misleading. Generally, I believe it's easier for the reader to not have the indirection via T, and the last column (FID) was anyways obtained for a specific T. So I'd recommend to substitute T with the actual value and provide throughput/latency actually per image in the final manuscript.

---

> ### Author Response · Authors · 2024-12-02
>
> Thank you for the comment. In Table 3, we use the same number of sampling steps per section. We will revise Table 3 accordingly in our final manuscript.
>
> Thank you again for taking the time to review our paper and for providing constructive feedback!

---

### Official Review · Reviewer_9DQC · 2024-11-06

**Soundness:** 4
**Presentation:** 4
**Contribution:** 3
**Rating:** 8
**Confidence:** 4

**Summary:**

The paper presents a new type of autoencoder for high resolution generative diffusion models. Autoencoders are used in generative diffusion models primarily to reduce the computation necessary for training and evaluation. However, there is some loss of representation capacity in mapping the image to the latent domain. The loss of representation capacity becomes more severe with higher downsampling ratios, which limits compute gains beyond those provided by 8X downsampling, as well as high-resolution generation.

The present paper aims to expand the possible range of downsampling ratios to as high as 64X. The paper achieves this first by analyzing the challenges of higher downsampling ratios, finding that staged architectures actually have worse performance than architectures with simple space-to-channel operations. Following this, the paper proposes a new architecture with residual space-to-channel and channel-to-space connections that effectively skip the information across network blocks, improving optimization High-resolution performance after adopting the residual space-to-channel operations still left some to be desired, so the paper further proposes a decoupled high-resolution adaptation procedure that finetunes the head of the decoder while finetuning the rest of the model. A second stage tunes the middle layers.

In numerical experiments the trained autoencoders exhibit better reconstruction accuracy than their counterparts from stable diffusion, which are commonly used in the community. The full diffusion models show promising FID numbers vs. stable diffusion with more efficient training in both quantitative and qualitative examples.

**Strengths:**

1. The paper is well-written and easy to follow.
2. The baselines using stable diffusion are well-known and recognized by the community.
3. The qualitative examples seem compelling.
4. The advantages with using less compute could be of significant impact.

**Weaknesses:**

1. The authors mostly use stable diffusion for all experiments, although several datasets are considered.
2. A more detailed quantitative analysis of why standard staged training has difficulties with gradient propagation is not presented.
3. The proposed modifications are somewhat incremental - this seems close to simply removing some skip connection convolutions + a reshape operation in the standard stable diffusion autoencoder. The extra fine-tuning procedure is more detailed, but is only necessary due to the autoencoder changes.
4. Autoencoder comparisons in Table 2 are only done at high downsampling ratios - not sure how the proposed architecture performs with shallow downsamples.
5. Only the FID metric (presumably with the Inception V3 backbone) is used to evaluate the models.

**Questions:**

My overall rating is due to the somewhat limited set of modifications and the limited experiments, which should be a topic for rebuttal. In addition, I also have the following questions:

1. Did you adopt all the same loss functions and GAN architectures as in the stable diffusion paper?
2. How often does the proposed architecture output artifacts?
3. Are the images in all figures random-samples or selected? This should be made clear in the text.
4. Did you consider other operations for space-to-channel other than averaging? Same question for channel duplication in the upsampling process.

---

> ### Author Response · Authors · 2024-11-25
> **Author Response**
>
> ### Q1: Novelty and Technical Contribution
> DC-AE is the first study to solve a critical and novel problem: addressing the degradation of reconstruction accuracy in high spatial-compression autoencoders. For the first time, DC-AE makes high spatial-compression autoencoders' reconstruction accuracy feasible for usage in latent diffusion models. It can benefit a large community using autoencoders, making high-resolution diffusion models more accessible as a research topic.
>
> DC-AE also provides novel insights into the challenges of increasing the spatial compression ratio of autoencoders and introduces novel designs to tackle these challenges. In addition, we believe keeping the solution simple is a strength of DC-AE instead of a weakness, as simple designs are usually easier to use and more general than complicated designs.
>
> ### Q2: Comparisons with More Autoencoders
>
> Thanks for the suggestion. We have updated Table 3 and Table 4, adding comparisons with Flux-VAE, SD-XL-VAE, SD3-VAE, and Asymmetric-VAE. **DC-AE delivers consistent and significant efficiency improvements over these autoencoders while maintaining similar or better generation performance.**
>
> ### Q3: Autoencoder Comparisons Under Low Spatial-Compression Ratio
>
> We have added reconstruction results under the low spatial-compression ratio setting in **Appendix I (Table 8)**. DC-AE delivers slightly better results than SD-VAE under this setting.
>
> | f8c4   | rFID | PSNR  | SSIM | LPIPS |
> | ------------------- | :--: | :---: | :--: | :---: |
> | SD-VAE |	0.63 | 24.99 | 0.71 | 0.063 |
> | DC-AE  | **0.46** |  **25.46** |  **0.73** | **0.057** |
>
> ### Q4: More Evaluation Metrics in Diffusion Experiments
>
> Thanks for the suggestion. In **Appendix J (Table 9)**, we have added a comprehensive evaluation with different evaluation metrics, including FID, inception score (IS), precision, recall, and CMMD. **Our DC-AE consistently delivers significant efficiency improvements while maintaining similar or better generation performance under different evaluation metrics.**
>
> ### Q5: Analysis of Why Standard Staged Training Has Optimization Difficulty
> We have added the loss landscape comparison between DC-AE and SD-VAE in **Appendix H (Figure 12)**. We find that DC-AE's loss landscape is flatter than SD-VAE's, indicating that it is easier for DC-AE to reach low-loss regions than SD-VAE. Since our work is not a theory paper, further theoretical analysis and proof are out of the scope of this paper. We leave it to future research.
>
> ### Q6: Loss Function and GAN Architecture
>
> We adopt the same loss functions and the same GAN architecture as in the stable diffusion paper.
>
> ### Q7: How often does the proposed architecture output artifacts?
>
> We have never observed our models outputting artifacts.
>
> ### Q8: Are the images in all figures random-samples or selected?
>
> Following the common practice, we select representative images for visualization in the figures. We have updated the captions to make it clear. In addition, we have added random image generation samples in Figure 16 and Figure 17.
>
> ### Q9: Did you consider other operations other than averaging/duplication?
>
> We have tried using 1x1 conv instead of averaging/duplication. It leads to worse results.
>
> | ImageNet 256x256    | rFID | PSNR  | SSIM | LPIPS |
> | ------------------- | :--: | :---: | :--: | :---: |
> | DC-AE with 1x1 conv |	1.06 | 22.86 | 0.62 | 0.094 |
> | DC-AE               | **0.69** | **23.85** | **0.66** | **0.082** |

---

> > ### Comment · Reviewer_9DQC · 2024-11-26
> >
> > I have read the other reviewers' reviews as well as the authors' reply to the reviews. I would like to thank the authors for their reply to my comments and I have correspondingly increased my score. My only remaining thought is that I find the claim "We have never observed our models outputting artifacts" dubious - all generative models occasionally produce artifacts.

---

> > > ### Author Response · Authors · 2024-11-26
> > > **Official Comment by Authors**
> > >
> > > Thank you for the prompt response! Regarding the artifacts, we mean **we have not observed our reconstruction models outputting artifacts.** We sincerely apologize for the misunderstanding. As for the artifacts from generation models, it depends on the diffusion models used.
> > >
> > > Thank you again for taking the time to review our paper and for providing constructive feedback!

---

### Official Review · Reviewer_rPFh · 2024-11-07

**Soundness:** 3
**Presentation:** 4
**Contribution:** 3
**Rating:** 8
**Confidence:** 2

**Summary:**

This paper looks at the problem of scaling image autoencoders, used with generative models, to higher spatial compression rates without reducing visual quality. The authors identify a problem with existing AEs: visual quality degrades with higher spatial compression rates (Fig. 2) even when increasing the number of channels in the latent representation to maintain a fixed total size (see the "latent shape" column in Table 2). They also observe a quality degradation when generalizing from low resolutions (used for training) to higher-resolutions.

These observations imply that "high spatial-compression autoencoders are more difficult to optimize" so the paper introduces an architecture change ("residual autoencoding") and a training change ("decoupled high-resolution adaptation") to improve results with high spatial compression rates and to reduce generalization error across resolutions.

Residual autoencoding (Fig. 4) replaces strided conv blocks used for up/downscaling with residual blocks that learn a residual added to space-to-depth (for downscaling) or space-to-depth (for upscaling) blocks. The model uses channel averaging to adjust the number of channels when there is a mismatch, and authors also use channel averaging for a new kind of residual block even when there is no up/downscaling.

Decoupled high-resolution adaptation (DHRA) switched from a single-stage approach for training the AE to a three-stage appraoch (Fig. 6). In the standard approach, the AE is trained on low-res data using all reconstruction losses (typically mse + perceptual + adversarial). In DHRA, the stages are:
 1. low-res images optimized without the adversarial loss
 2. finetune with high-res images with only the "inner" layers (end of encoder, beginning of the decoder) trained. Other layers are frozen and, again, no adversarial loss is used.
 3. finetune again with reconstruction and adversarial losses. Only the final layers of the decoder are trained.

The authors argue that this multi-stage approach is more efficient than training the full model on high-res images, and it prevents the adversarial loss from affecting the encoder which helps with generalization.

The authors validate their claims by training models and evaluating reconstruction quality (Table 2) and generation quality (Tables 3 and 4). They also provide some example images for qualitative evaluation (Fig. 7 and in the supplementary material).

**Strengths:**

The primary strength of this paper is the development of an effective and relatively simple solution to a real-world deficiency that impacts all research and products using autoencoders. Specifically, the use of residual autoencoding and DHRA allow either better visual quality (e.g., in Table 3 look at rows witht he same number of tokens like SD-VAE-f16 with patch-size=2 or SD-VAE-f32 with patch-size=1 to DC-AE-f32) or similar visual quality with lower latency and higher throughput (e.g., Table 3 comparing SD-VAE-f32 to DC-AE-f64).

Another strength is that the authors evaluate their method on both reconstruction and generation problems. Improvements on just reconstruction still have some value, but many improvements to reconstruction quality do not translate to gains on the generation side.

**Weaknesses:**

A more detailed summary of the DC-AE architecture would help. Fig. 4 provides a high-level overview, but I'm not sure how literal or complete it is, and the details of the "Encoder/Decoder Stages" are not provided. To be fair, the authors do provide code, but architecture and training details should be in the paper (probably in the appendix).

The conjecture in Section 3.3 that DC-AE helps because otherwise the "diffusion model needs to simultaneously learn denoising and token compression when using a patch size > 1" is pretty hand-wavy. For example, I can always set patch_size=1 and add add a space-to-depth layer to the encoder and claim that now the encoder alone is responsible for token compression. That said, I don't have a better explanation, but I think it's ok for papers to accept that the contribution is a better architecture justified with empirical evaluation.

**Questions:**

Regarding generalization across resolutions, I'm wondering how much changes to the training procedure can close the gap even with single-stage training. In particular, it makes sense that the statistics for a fixed-size patch (say 32x32 pixels) is very different between a 256x256 image and 1024x1024 pixel image of the same scene. You could train on 256x256 *crops* from a 1024x1024 dataset (rather than downscaling), but then you'd have a generalization gap in the other direction.

In other work, researchers try to avoid this problem by generating low-res training sets by randomly downscaling high-res sets. With enough training, this approach lets the model see the full range of statistics. Was this done for the experiments in the DC-AE paper? If not, do you think it's sufficient or perhaps complementary to DHRA? Either way, I still see potential benefits of DHRA, e.g., maybe it's good that the GAN doesn't influence the encoder (and thus the latent space), but this isn't investigated, in isolation, in the paper.

You can also frame higher spatial-compression rates as meaning that the model sees less training data (as measured by number of latents) per fixed-size image. This means that any negative affects from the boundary are exacerbated (e.g., if zero-padding is used for conv layers). In the extreme, imagine using f256 with 256x256 training data -- we certainly wouldn't expect this model to generalize to higher resolutions. Some discussion of these effects would help, and investigation of whether the generalization issue is mostly one of different pixel statistics or of boundary artifacts would strengthen the paper.

Did you experiment training different numbers of layers in Stage 2 and Stage 3? A graph showing the impact (training speed vs. evaluation metrics) as the number of layers increases would be interesting.

---

> ### Author Response · Authors · 2024-11-25
> **Author Response**
>
> ### Q1: Investigations on the Generalization Gap
> We conduct ablation study experiments to investigate the generalization gap and summarize the results in **Appendix F (Table 7)**. We find training with image crops leads to worse results than training with downscaled images in our case. Training with 50% downscaled images and 50% image crops can improve the rFID from 7.4 to 2.7. It shows that this strategy can partially address the generalization gap. Based on this result, we conjecture combining this strategy with our decoupled high-resolution adaptation may further boost the performance. We leave it to future research.
>
> | Method | Downscale | Crop | 50% Downscale, 50% Crop | Decoupled High-Resolution Adaptation |
> |--------|:---------:|:----:|:-----------------------:|:------------------------------------:|
> |  rFID  |    7.40   | 8.32 |           2.70          |                **0.18**              |
>
> By the way, we want to cite some related works (e.g., training with image crops, training with randomly downscaling high-res sets) in **Appendix F**. We wonder if the reviewer has any suggestions on which papers we should cite in this part. We greatly appreciate your feedback.
>
> ### Q2: Ablation Study on Training Different Numbers of Layers in Stage 2 and Stage 3
> We have added ablation study results in **Appendix G (Figure 11)**. Thanks for the suggestion.
>
> ### Q3: Detailed Summary of DC-AE Architecture and Training Details
> We have added a detailed summary of DC-AE architecture and training details in **Appendix E**.
>
> ### Q4: Conjecture in Section 3.3
> Thank you for the suggestion. We have revised Section 3.3 accordingly.

---

> > ### Comment · Reviewer_rPFh · 2024-12-02
> > **citation for random downscaling**
> >
> > > By the way, we want to cite some related works (e.g., training with image crops, training with randomly downscaling high-res sets) in Appendix F. We wonder if the reviewer has any suggestions on which papers we should cite in this part. We greatly appreciate your feedback.
> >
> > Probably not the first paper to do it, but Section 6.1 in this 2017 ICLR paper talks about randomized downscaling for training an image autoencoder:
> >
> > https://arxiv.org/abs/1611.01704
> >
> > (note that I am not an author on this paper)

---

### Official Review · Reviewer_jBHL · 2024-11-07

**Soundness:** 2
**Presentation:** 2
**Contribution:** 2
**Rating:** 6
**Confidence:** 4

**Summary:**

This paper presents a family of autoencoders that achieve comparable or better reconstructions compared to the SD autoencoders while having a much stronger spatial compression rate, resulting in efficiency gains for downstream tasks such as latent diffusion training.
Using the techniques or residual autoencoding and decoupling high-resolution adaptation, the model is able to generalize well to higher resolutions while having a very large compression factor (x32, 64 vs 8 for SD-VAE).

**Strengths:**

* The paper is well written and easy to follow.
* The method allows to significantly reduce the computational requirements to train large scale diffusion models at high resolution, making it more accessible as a research topic.
* Evaluation of multiple metrics across 4 different datasets are reported.
* Evaluations on downstream diffusion training is also provided.
* Qualitative examples clearly showcase the improvements brought about by the proposed model.

**Weaknesses:**

* What parameters were used for sampling the diffusion models (sampler, number of sampling steps, guidance scale) ? A more through investigation on the impact on sampling quality would be useful to get a better grasp of the limitations of this method.
* Missing ablations on the constituent parts of DC-AE.
* In tables 3 and 4, missing comparisons with more recent autoencoders such as the one from SD-XL, SD3, and Asymetric Autoencoder.
* In table 5. the PixArt model trained with DC-AE achieves a lower CLIP score than the one trained with SD-VAE while the text says otherwise (maybe a typo?). What if the memory usage is equalized with the SD-VAE by increasing the batch size, do the improvements get larger or do they stagnate ?

**Questions:**

* While increasing the patch size is one method to reduce spatial resolution, one can also use a pixel shuffle operation in order to achieve a lower spatial dimension with a higher channel dimension. How does this simple operation, using SD-VAE compare with your method ?
* Training LDMs requires a shift and scale factor to be applied to the latents before the diffusion process, how are these values computed in your experiments ?

---

> ### Author Response · Authors · 2024-11-25
> **Author Response**
>
> ### Q1: Diffusion Sampling Hyperparameters
>
> We have added details of the diffusion sampling hyperparameters in **Appendix B**. For the DiT models, we use the DDPM sampler with 250 sampling steps and a guidance scale of 1.3. For the UViT models, we use the DPMSolver sampler with 30 sampling steps and a guidance scale of 1.5.
>
> We also conduct ablation study experiments on the diffusion sampling hyperparameters and summarize the results in **Appendix B (Figure 9)**. **DC-AE provides significant speedup over the baseline models while maintaining similar or better generation performance under different diffusion sampling hyperparameters.**
>
> ### Q2: Ablation Study Experiments on DC-AE
>
> We have added ablation study experiments on DC-AE in **Appendix C (Table 6)**. Both residual autoencoding and decoupled high-resolution adaptation contribute significantly to DC-AE's superior performances in high spatial-compression settings.
>
> | Decoupled High-Resolution Adaptation | Residual Autoencoding | rFID  | PSNR  | SSIM | LPIPS |
> |:------------------------------------:|:---------------------:|:-----:|:-----:|:----:|:-----:|
> |                                      |                       | 16.84 | 19.49 | 0.48 | 0.282 |
> |               &check;                |                       | 5.54  | 21.13 | 0.54 | 0.228 |
> |               &check;                |         &check;       | **0.22**  | **26.15** | **0.71** | **0.080** |
>
> ### Q3: Comparisons with Recent Autoencoders in Table 3 and Table 4
> We have updated Table 3 and Table 4, adding comparisons with Flux-VAE, SD-XL-VAE, SD3-VAE, and Asymmetric-VAE. **DC-AE delivers consistent and significant efficiency improvements over these autoencoders while maintaining similar or better generation performance.**
>
> ### Q4: Typo in Table 5
> Thanks for pointing out the typo. We have fixed it in Table 5.
>
> ### Q5: Impact of Increasing the Batch Size
> We use the same training setting as the baseline for a fair comparison. In practice, users can increase the batch size when training diffusion models with our DC-AE under the same memory budget, potentially leading to better generation results. Thanks for the suggestion.
> | Model | FID (w/o CFG) | FID (w/ CFG) |
> |-------|:-------------:|:------------:|
> | DC-AE-f32 + DiT-XL (original batch size) | 9.56 | 2.84 |
> | DC-AE-f32 + DiT-XL (larger batch size) | **7.45** | **2.50** |
>
> ### Q6: Comparison to SD-VAE + PixelUnshuffle (Space-to-Channel)
> Combining PixelUnshuffle (Space-to-Channel) with SD-VAE is mathematically equivalent to increasing the patch size. DC-AE clearly outperforms this strategy, as shown in our experiments.
>
> ### Q7: Latent Scaling and Shifting Factors
> We have added detailed discussions in **Appendix D**. Given a dataset, we compute the root mean square of the latent features and use its multiplicative inverse as the scaling factor for our autoencoders. We do not use the shifting factor for our autoencoders.

---

### Author Response · Authors · 2024-11-25
**General Response**

We sincerely thank all reviewers for their constructive comments. We are encouraged to see that the reviewers recognize our contributions to accelerating high-resolution diffusion models. We have updated our manuscript accordingly (highlighted in gold color). We summarize the key updates of our revision below:
- We updated Table 3 and Table 4, adding comparisons with recent autoencoders including Flux-VAE, SD-XL-VAE, SD3-VAE, and Asymmetric-VAE. DC-AE delivers consistent and significant efficiency improvements over these autoencoders while maintaining similar or better generation performance.
- We added ablation study experiments on diffusion sampling hyperparameters in Appendix B. DC-AE provides significant speedup over the baseline models while achieving similar or better generation performance under different diffusion sampling hyperparameters.
- We added ablation study experiments on DC-AE in Appendix C.
- We added investigations on the generalization gap in Appendix F.
- We added a loss landscape comparison in Appendix H.
- We added image reconstruction results under the low spatial-compression ratio setting in Appendix I.
- We added image generation results with more evaluation metrics in Appendix J. DC-AE consistently delivers significant efficiency improvements while delivering similar or better generation results under different evaluation metrics.
- We added model scaling results in Appendix K. DC-AE benefits more from scaling up the diffusion model than SD-VAE.

Please see our detailed responses below each review. We greatly appreciate your feedback and welcome any additional questions.

---

### Meta-Review · Area_Chair_h6Jw · 2024-12-18

**Metareview:**

This paper explores image auto-encoders with a high spatial compression rate in the context of image generation. The proposed model results in efficiency gains while maintaining downstream performance. The manuscript was reviewed by four knowledgeable reviewers who acknowledged that the paper was well written and easy to follow (jBHL, 9DQC), the proposed solution was relatively simple, sound and effective (rPFh, pFgo), offering significant speedups (pFgo) and reduction of resources required to train diffusion models (jBHL), highlighting the potential of the proposed approach (9DQC).

The reviewers's main concerns were:
1. Missing ablations (jBHL) and analyses on the generalization across resolutions (rPFh)
2. Missing architecture details (rPFh)
3. Proposed modifications somewhat incremental (9DQC), and no theoretical analyses or toy examples highlighting the potential of the proposed design (iQFp)
4. Only FID metric used for evaluation (9DQC)

During rebuttal and discussion, the authors emphasized novelty and technical contribution, added comparisons with more auto-encoders and ablations, as well as metrics (PR, IS, CMMD), shared additional details on the method and experimental setup, and scaled some experiments. After discussion, the reviewers unanimously recommend acceptance. The MR agrees with the reviewers' assessment and recommends to accept.

**Additional Comments On Reviewer Discussion:**

See above.

---

### Decision · Program_Chairs · 2025-01-22

Accept (Poster)